# The unprecedented 2015/16 Tasman Sea marine heatwave

Eric C.J. Oliver[1,2], Jessica A. Benthuysen[3], Nathaniel L. Bindoff[1,2,4], Alistair J. Hobday[5], Neil J. Holbrook[1,2], Craig N. Mundy[1] & Sarah E. Perkins-Kirkpatrick[6,7]

The Tasman Sea off southeast Australia exhibited its longest and most intense marine heatwave ever recorded in 2015/16. Here we report on several inter-related aspects of this event: observed characteristics, physical drivers, ecological impacts and the role of climate change. This marine heatwave lasted for 251 days reaching a maximum intensity of 2.9 °C above climatology. The anomalous warming is dominated by anomalous convergence of heat linked to the southward flowing East Australian Current. Ecosystem impacts range from new disease outbreaks in farmed shellfish, mortality of wild molluscs and out-of-range species observations. Global climate models indicate it is very likely to be that the occurrence of an extreme warming event of this duration or intensity in this region is respectively ≥330 times and ≥6.8 times as likely to be due to the influence of anthropogenic climate change. Climate projections indicate that event likelihoods will increase in the future, due to increasing anthropogenic influences.

[1] Institute for Marine and Antarctic Studies, University of Tasmania, 20 Castray Esplanade, Battery Point, Private Bag 129, Hobart, Tasmania 7001, Australia. [2] Australian Research Council Centre of Excellence for Climate System Science, University of Tasmania, Private Bag 129, Hobart, Tasmania 7001, Australia. [3] Australian Institute of Marine Science, PMB 3, Townsville MC, Townsville, Queensland 4810, Australia. [4] Antarctic and Climate Ecosystem Cooperative Research Centre, University of Tasmania, Private Bag 80, Hobart, Tasmania 7001, Australia. [5] CSIRO Oceans and Atmosphere, Hobart, Tasmania 7000, Australia. [6] Climate Change Research Centre, University of New South Wales, Gate 11 Botany Street, Library Walk, Level 4, Mathews Building, Sydney, New South Wales 2052, Australia. [7] Australian Research Council Centre of Excellence for Climate System Science, University of New South Wales, Gate 11 Botany Street, Library Walk, Level 4, Mathews Building, Sydney, New South Wales 2052, Australia. Correspondence and requests for materials should be addressed to E.C.J.O. (email: eric.oliver@utas.edu.au).

Recent indications are that the frequency of extreme warming events in the ocean is increasing globally[1]. In both 2015 and 2016, approximately one quarter of the ocean surface area experienced a marine heatwave (MHW; based on the definition by Hobday et al.[2]) that was either the longest or most intense ever recorded since global satellite records began in 1982 (Supplementary Fig. 1). These events have devastated marine ecosystems globally but there is limited understanding of their physical drivers and the role of anthropogenic climate change. Individual MHWs have been examined in terms of their definition[2], physical drivers[3–7] and ecological impacts[7–11] and inferences have been made to the role of climate change[9–12].

During the austral summer of 2015/16, sea surface temperatures (SSTs) off southeast Australia were up to 3–4 °C above climatological averages, the warmest on record for that region. At this time, temperature anomalies $>1$ °C were contiguous over an area nearly 21 times the size of Tasmania ($1.4 \times 10^6$ km$^2$), anomalies $>2$ °C over an area more than seven times the size of Tasmania ($4.8 \times 10^5$ km$^2$) and anomalies $>3$ °C over an area nearly half the size of Tasmania ($3.2 \times 10^4$ km$^2$). This event impacted regional biodiversity, such as the appearance of marine species normally found further north, and was a detrimental stressor on coastal fishery and aquaculture industries, including the abalone, Pacific oyster and Atlantic salmon industries. Even human interactions with the ocean were modified, where swimmers and surfers noted the unusual warmth of the waters around Tasmania; a region normally noted for its relatively cold waters for swimming.

Marine ecosystems are strongly influenced by extreme climatic events including heatwaves[8,9], cold snaps[13], storms[14] and floods[15,16]. MHWs, which can be caused by a combination of atmospheric and oceanographic processes, have led to a range of ecological impacts, including mass mortality of abalone (off Western Australia[17]), benthic habitat loss (Mediterranean Sea[8]) and altered human use of the ocean (that is, fisheries; northwest Atlantic and off Western Australia[10,11]). In the coastal waters off eastern Tasmania, a complete die-off of giant kelp (*Macrocystis pyrifera*) was reported during a warm weather event in 1988 (ref. 18) and may have been associated with a MHW.

The ocean off southeastern Australia is a global warming hotspot[19]. The near-surface waters there are warming at nearly four times the global average rate[20,21] and these increasing temperatures are seen as deep as 750 m[22]. This warming has been linked to enhanced southward transport in the East Australian Current, driven by increased wind stress curl across the mid-latitude South Pacific[21,23]. The extension of this current south of *ca*. 33°S consists of an unsteady train of mesoscale eddies, resulting in increased eddy mixing within the Tasman Sea[24]. Future projections under anthropogenic climate change indicate continued strengthening of the southward transport in the East Australian Current Extension, linked with increased wind stress curl over the South Pacific, and a corresponding increase in the likelihood of extreme temperature events[25–28].

This study discusses the 2015/16 Tasman Sea MHW from observations and ocean models, diagnoses its physical drivers and the role of anthropogenic climate change, and describes the ecological impacts that occurred. We investigate the hypotheses that the MHW in the Tasman Sea during austral summer 2015/16 was driven by anomalous southward transport in the East Australian Current and that anthropogenic climate change increased the likelihood of such an event. Our approach involves a synthesis of observations, theory and numerical models. First, we describe the event from remotely-sensed SST measurements as well as near-shore *in situ* sub-surface temperature and velocity measurements. Second, we determine the primary physical drivers of the MHW using ocean model estimates to determine the relative contributions

of horizontal temperature advection and air–sea heat fluxes. Third, we use global climate models to estimate the increased risk of MHWs in the Tasman Sea, with the duration and intensity of this particular event observed in the summer of 2015/16, due to anthropogenic climate change. Finally, we document how the 2015/16 MHW affected regional coastal ecosystems, including the billion dollar aquaculture and fisheries industries.

## Results

**Characteristics of the 2015/16 Tasman Sea MHW.** Time series of SST, spatially averaged over the region offshore of southeast Australia (hereafter the SEAus region: bounded by (147°E, 155°E) and (45°S, 37°S); Fig. 1a, black box), show that the summer of 2015/16 was much warmer than recent summers (Fig. 1b). The regionally averaged SST anomalies—both daily, remotely sensed (Fig. 1c, black line) and monthly, *in situ* based (Fig. 1c, circles)— were $>1$ °C warmer than average for the entire period from early September 2015 to May 2016, and 1.5–3 °C warmer for the period from November 2015 to February 2016. The monthly mean temperatures at this time were also both the warmest absolute temperatures and the warmest anomalies on record since 1880 (Supplementary Fig. 2).

Applying the Hobday et al.[2] MHW definition (see Methods) to daily, remotely sensed SSTs, we found that the SEAus region was in a continuous MHW state from 9 September 2015 to 16 May 2016 (Fig. 1b,c, red shaded area). According to the MHW metrics, this event had a duration of 251 days, a maximum intensity of 2.9 °C, a mean intensity of 1.8 °C and a cumulative intensity of 443 °C-days. By all four metrics, this event was the largest on record (Fig. 1d,e and Supplementary Fig. 3). Monthly SST data (Hadley Centre Sea Ice and Sea Surface Temperature, HadISST) showed that 6 of the 9 months corresponding to the event (September 2015–May 2016) had among the top ten largest magnitude monthly anomalies on record since 1900 (red-filled circles in Fig. 1b,c and Supplementary Fig. 2). In addition, this event consisted of the largest magnitude 9-month running mean SST anomalies since 1900. In summary, this event was both the longest and the most intense MHW on record in this region.

**Near-shore observations of the event.** Time series of temperature at a number of monitoring sites around Tasmania situated between 6 and 19 m depth (Fig. 2a–m) indicated that the summer of 2015/2016 was the warmest on record at most sites since measurements began in 2004. The January averaged temperature (anomaly) at the Australian National Mooring Network site at Maria Island (42.59°S,148.23°E) was 18.4 °C (+2.02 °C), which was more than 1 °C warmer than the previous record summer of 2011/12 (Fig. 2g). The 136-day period from 30 December 2015 to 13 May 2016 was the longest and most intense MHW on record at Maria Island (a maximum (mean) intensity of +3.2 °C (+1.8 °C); Fig. 2g, red-shaded region and blue line). Interestingly, at Maria Island record warm temperature anomalies were measured at 85 m depth during February 2016 but these anomalies were interspersed with very cool periods (not shown), indicating that the persistence of this event was likely confined to shallower depths, at least near the coast. During the summer of 2015/16, the remaining sites also showed that either the most intense MHW on record occurred (+1–3 °C, Fig. 2a–m, red-shaded regions) and/or the longest MHW on record occurred (up to 125 days at Bicheno and Cape Peron, Fig. 2a–m, blue lines).

These anomalous temperatures may be attributed to regional changes in coastal circulation and anomalous southward transport of warm waters. Strong southward flow, broadly indicative of an intensified East Australian Current Extension, was recorded at Maria Island in December 2015 and January 2016 (Fig. 2n).

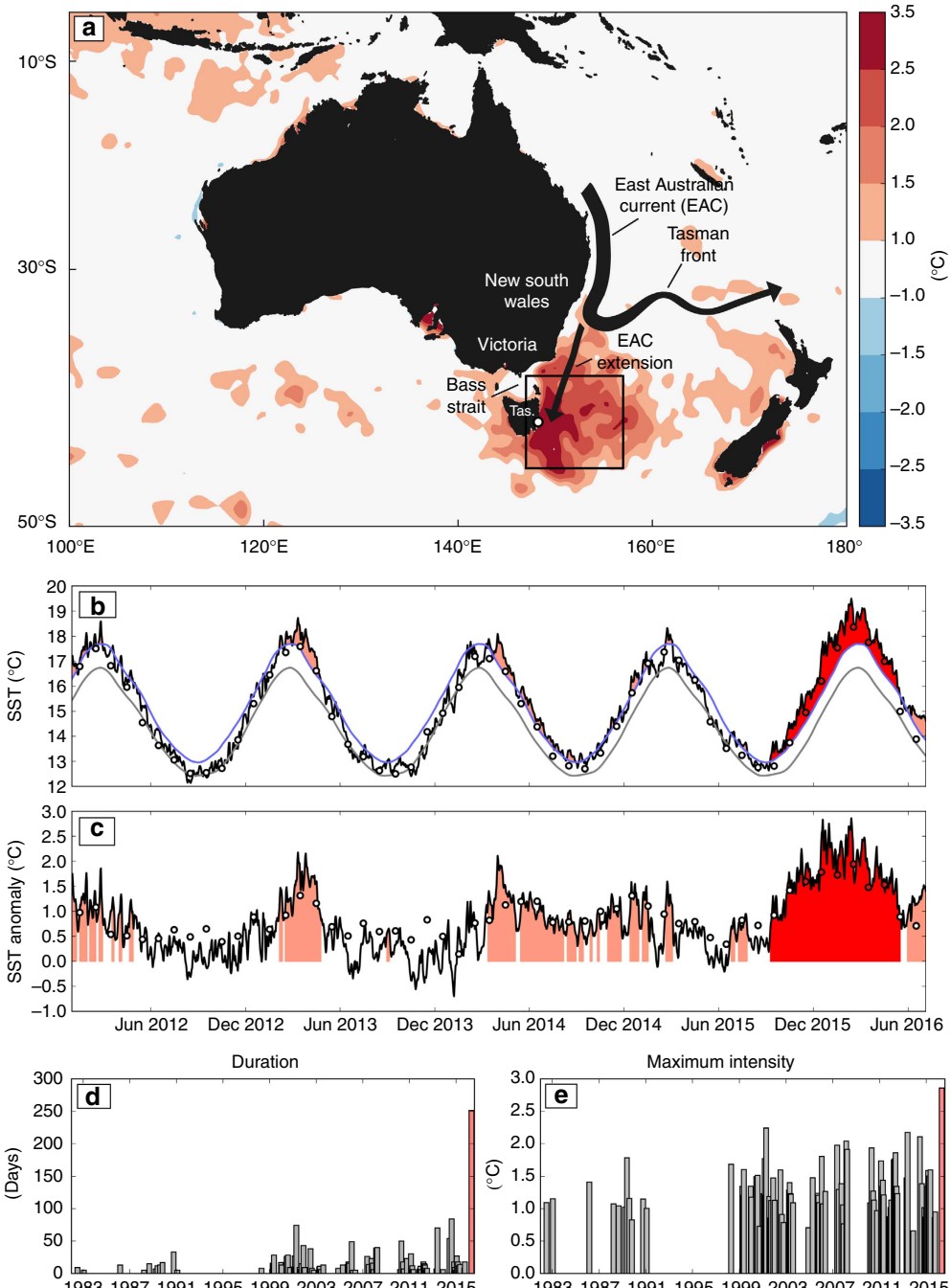

**Figure 1 | The 2015/16 MHW off southeast Australia. (a)** The mean 2015/16 austral summer (December to February) mean SST anomalies from NOAA OI SST, the box used to define the southeast Austraila (SEAus) region (black lines) and the location of the Maria Island Time Series (open circle). Anomalies are relative to the 1982–2005 climatology. Also shown are time series of (**b**) SST and (**c**) SST anomalies averaged over the SEAus region since 2012 from NOAA OI SST (black lines) and HadISST (circles). The red-filled circles in **b,c** indicate which months during the event were among the top ten on record since 1880. The grey and blue lines in **b** indicate the climatological mean and 90th percentile threshold, respectively, calculated from NOAA OI SST. The pink-shaded regions in **b,c** indicate all MHWs detected using the Hobday *et al.*[2] definition and the red-shaded region is the 2015/16 event. The (**d**) duration and (**e**) maximum intensity are shown for this event as red bars along with values for all previous events on record back to 1982.

In fact, the strongest December–January anomalies of depth-averaged southward flow on record occurred in 2015/16 ($-2.48 \, \text{cm} \, \text{s}^{-1}$) and the summer of 2015/16 was the only summer on record with average December–January flow in the southward direction at Maria Island.

**Evolution of the MHW.** The MHW's rise, peak and decay were examined from September 2015 through to May 2016 using monthly mean SST anomalies (Fig. 3). This was supplemented by examining potential forcing mechanisms based on concurrent sea surface currents (Fig. 4) and surface air temperature anomalies and winds (Fig. 5). There is a clear indication of the event having propagated southward over the domain and coinciding with strong southward surface flow and moderately warm air temperatures.

The rise of the MHW occurred from September to November 2015. In September 2015, there were warm anomalies ( + 2–3 °C)

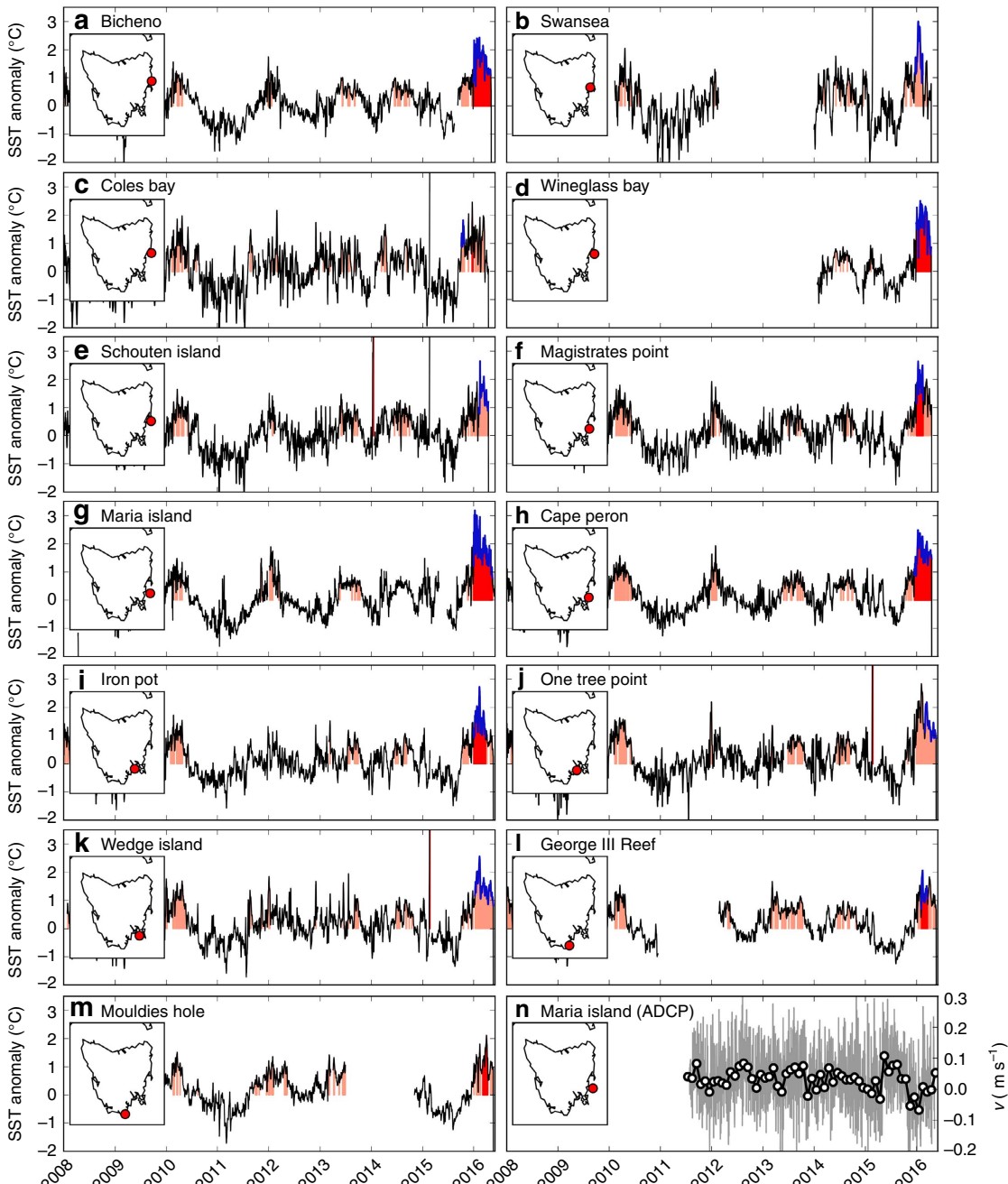

**Figure 2 | Near-shore temperature and velocity records around Tasmania. (a–m)** Temperature anomalies are shown for 13 sites along Tasmania's eastern shelf. The shaded regions indicate each MHW detected using the Hobday *et al.*[2] definition, the red-shaded region is the largest event (by maximum intensity) and the blue line indicates the longest event. (**n**) Depth-averaged meridional velocity measured with an Acoustic Doppler current profiler (ADCP) at the Maria Island Australian National Mooring location (daily means as grey line, monthly means as open circles).

off the New South Wales south coast that extended zonally far into the Tasman Sea (Fig. 3a), although anomalies of this magnitude and spatial extent are not uncommon at this time of year. Through October and November, these warm anomalies intensified and shifted southward leading to a coherent area of +1.5–3 °C covering the entire east coast of Tasmania and north along the edge of Bass Strait to coastal Victoria; the warm anomalies extended offshore as far as 155°E (Fig. 3b,c).

Strong eddy activity developed between 36°S and 40°S, and by November southward surface currents were continuous over 5° of latitude linking the water masses off the southeastern portion of mainland Australia to northeastern Tasmania (Fig. 4a–c). In fact,

surface eddy kinetic energy in this region from September through March was higher in 2015/16 than the previous 3 years (Supplementary Fig. 4). Starting in October 2015, air temperatures were 0.5–2.5 °C warmer than average over the ocean off southeast Australia (Fig. 5b), but this weakened over the remainder of the summer. In addition, October experienced northwesterly winds off southeastern Australia, at a time when air over land was extremely warm and these winds could have blown the warm air out over the Tasman Sea (Fig. 5b).

The MHW peaked from December 2015 to March 2016. In December 2015, the warm anomalies intensified significantly, focused along a +3–4 °C core oriented meridionally over

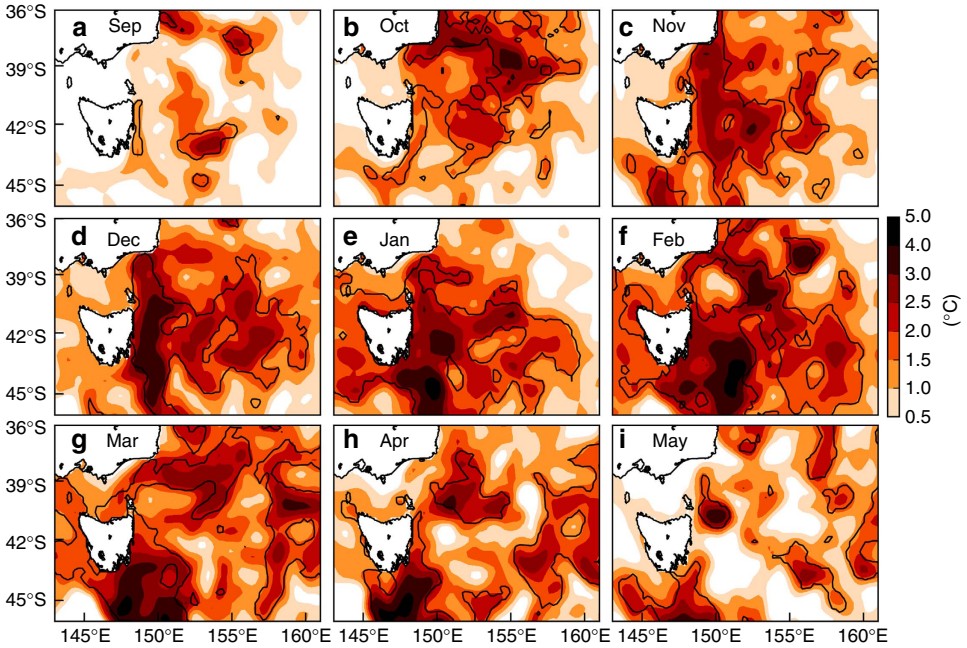

**Figure 3 | Monthly mean SST anomalies off southeastern Australia from September 2015 to May 2016.** Sea surface temperature data is NOAA OI SST. Only positive anomalies are shown, monthly for (**a-i**) Sep 2015 through May 2016; anomalies are relative to the 1982–2005 base period. Black contours enclose areas where at least 90% of the days in that month were part of an identified MHW.

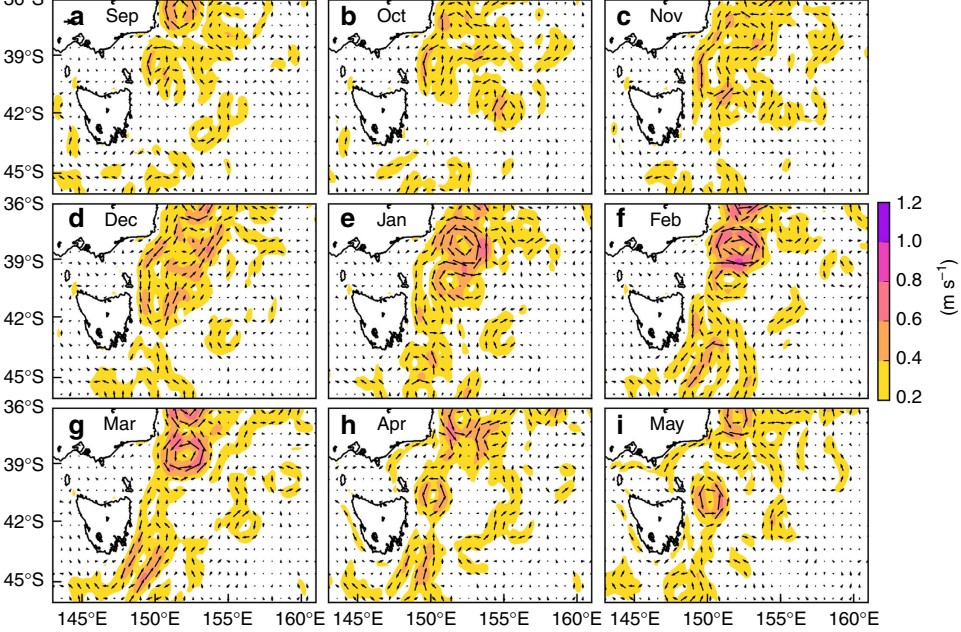

**Figure 4 | Monthly mean surface currents off southeastern Australia from September 2015 to May 2016.** Ocean current data is IMOS OceanCurrent. Currents are shown monthly for (**a-i**) Sep 2015 through May 2016. A reference arrow of 0.5 m s⁻¹ shown in upper left of (**a**) and colours indicate current speed.

45–40°S, sandwiched between the east coast of Tasmania and 150°E (Fig. 3d). Patches of warm anomalies up to +3 °C could be found as far east as 155–160°E. In January 2016, the southward shift of the entire warming pattern was apparent (Fig. 3e). Surface temperature anomalies north of 40°S were generally less than +1.5–2 °C and the warmest anomalies (4–5 °C) could be found as far south as 45°S. Strong southward flows persisted off eastern Tasmania in both December 2015 and January 2016 (Fig. 4d,e).

In February and March 2016, most of the domain bounded by 45–40°S and 145–160°E contained warm anomalies of at least +1 °C, with most >2 °C (Fig. 3f,g), and there were indications of further southward movement of the whole pattern with re-incursions of the warmest anomalies into southeastern Tasmania in March 2016.

The MHW decayed from April onwards. By then, most of the warm anomalies had moved out of the domain (Fig. 3h,i) and the

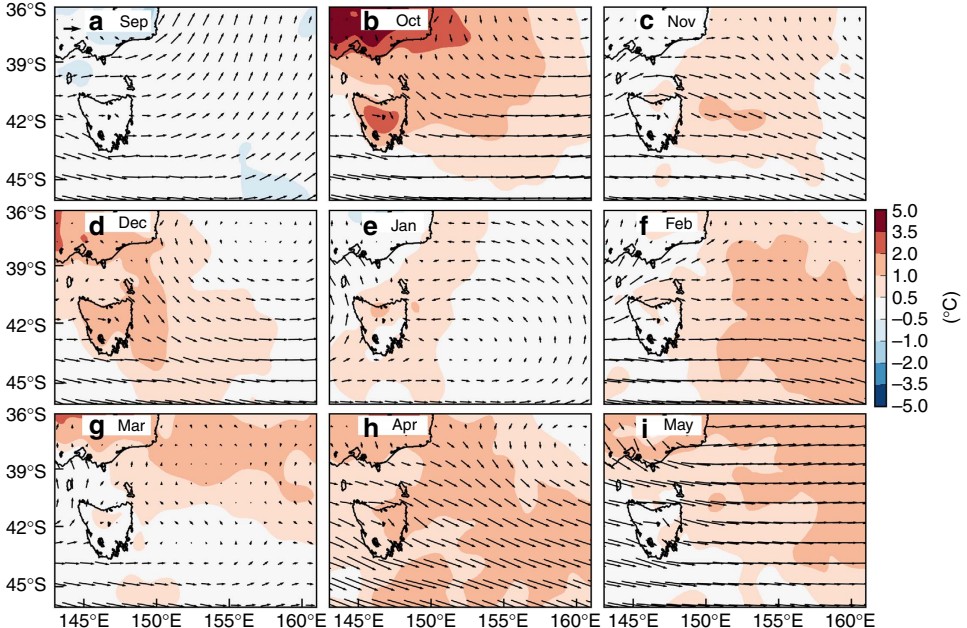

**Figure 5 | Monthly mean surface air temperature anomalies and surface winds off southeastern Australia from September 2015 to May 2016.** Atmospheric data is from NCEP CFSv2. Air temperatures and winds are shown monthly for (**a-i**) Sep 2015 through May 2016. Colours indicate air temperature anomalies and arrows indicate surface winds. Anomalies are relative to the 2012–2016 period. A reference arrow of 5 m s$^{-1}$ shown in upper left of **a**.

velocity field relaxed to its state consisting of a relatively incoherent set of independent eddies, with no longer a continuous southward flow linking a broad swath of southeast Australian coastline (Fig. 4h,i).

**Driving mechanisms.** We performed a temperature budget analysis to determine the relative roles of temperature advection and air-sea heat flux in causing the extreme warming off southeast Australia during the austral summer of 2015/16. The approach follows Benthuysen *et al.*[5] and Chen *et al.*[29], whereby the temperature budget (equation (1), Methods) is volume-averaged horizontally over the SEAus region and vertically down to 100 m depth, which captured most of the warming signal. For example, Argo profiles offshore of Tasmania show surface intensified warming with respect to climatology over the upper 100 m during the peak of the event (January/February 2016, for example, profile CS5904261_92 on 30 January 2016 (ref. 30)).

The temperature in the SEAus region was derived from OceanMAPS[31], and SSTs at 2.5 m depth compared well against observed SSTs (Supplementary Fig. 5, black and grey lines). Temperature anomalies from OceanMAPS exhibited similar warming down to 100 m depth, decaying down to 200 m beyond which the anomalies were relatively small at 350 and 500 m depths (Supplementary Fig. 5). Horizontal temperature advection was calculated from OceanMAPS velocities and temperatures, and the air–sea heat flux was derived from National Centers for Environmental Prediction (NCEP) Climate Forecast System Version 2 (CFSv2)[32,33] outputs.

The contributions of horizontal advection ($T_H$) and air-sea heat flux ($T_Q$) to the temperature budget were calculated along with the total volume-averaged temperature change ($T_V$) from 1 September for four consecutive September-March periods: 2012/ 13, 2013/14, 2014/15, and 2015/16. The remainder required to close the temperature budget was termed the residual. The climatological temperature budget, calculated over the first three periods, indicated that both horizontal advection and air-sea heat flux contribute strongly to the temperature during summer

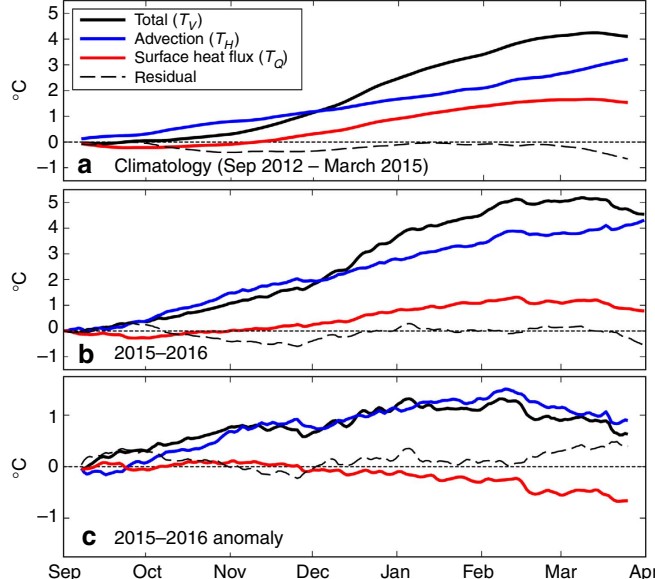

**Figure 6 | Temperature budget during the 2015/16 MHW.** (**a**) The climatology of the volume-averaged (0–100 m, SEAus region) temperature ($T_V$, black line) along with contributions due to horizontal advection ($T_H$, blue line), air-sea heat flux through the surface ($T_Q$, red line) and the residual (dashed line). The climatology was calculated over three consecutive September–March periods (2012/13, 2013/14, 2014/15). (**b**) The temperature budget, as in **a**, but for 2015/16 only. (**c**) The anomalous temperature budget during 2015–2016, calculated as the difference between **b**,**a**.

(Fig. 6a, blue, red and black lines), with a small residual (dashed line, typically less than 0.5 °C). Climatologically, by mid-February advection contributed ∼60% of the warming while air-sea heat flux contributed ∼40% of the warming.

In 2015/16 the volume-averaged temperature (Fig. 6b, black line) peaked in February and March at about 1 to 1.5 °C warmer

than climatology (Fig. 6c, black line). The contribution from advection was very strong for the entire period from October to March (Fig. 6b, blue line) and dominated the temperature budget over the air-sea heat flux (Fig. 6b, red line). By mid-February, advection contributed $\sim 80\%$ of the warming while $T_Q$ contributed $\sim 20\%$ of the warming (Fig. 6b). The contribution of air-sea heat flux was anomalously low over December to March (Fig. 6c, red line). This surprising result may be due to the anomalously warm SSTs already present, and thus the convergence of heat and anomalous warming (Fig. 6c, black and blue lines) came entirely from the advection term. Therefore, the warming associated with the 2015/16 MHW was primarily driven by anomalous temperature advection into the SEAus region. This conclusion of the primary source of heat from advection also holds if we use monthly air-sea heat fluxes from the Global Ocean Data Assimilation System (GODAS)[34,35] over the same time period (Supplementary Fig. 6).

We have decomposed the horizontal advection component into four sub-components: the temperature budget contributions from advection across the north, south, west and east faces of the SEAus box. In 2015/16, there was anomalously strong temperature advection inward (positive) across the north face (solid blue line, Supplementary Fig. 7). There was also anomalously strong temperature advection outward (negative) across the west face (solid red line) but not enough to compensate for the input across the north face—leading to an imbalance of 1–2 °C by February–March 2016. The anomalous input/output across the south (dashed blue line) and east (dashed red line) faces were in near-balance throughout 2015/16. We directly associate the input across the north face with the EAC Extension and therefore claim that the MHW was driven by an anomalous EAC Extension event.

The 2015/16 Tasman Sea MHW occurred during one of the largest El Niño events on record. While previous research has demonstrated that the time signature of El Niño—Southern Oscillation (ENSO) variability is evident in Tasman Sea temperatures, nutrients and biota[20,36–41], and sea level as decadal ENSO variations[42], the signals are relatively weak[43,44] with ENSO SST variations likely to be due mostly to regional air–sea interaction[43,45,46]. Dynamically, the main ocean waveguide for ENSO sea-level variations is via the Indonesian Archipelago and poleward along the west coast of continental Australia[47,48], rather than Australia's east coast. Thus, ENSO appears to play a weak to modest role in East Australian Current transport variations[49], via the influence of Rossby waves from offshore[50]. Thus, although the El Niño event of 2015/16 was indeed very large, there are no reports in the published literature to suggest that the unprecedented magnitude and/or duration of the Tasman Sea marine heatwave could be due solely to this event.

**The role of anthropogenic climate change.** Long-term ocean warming has been occurring concurrently with year-to-year variations in ocean circulation and air–sea heat fluxes. Ongoing warming might be expected to raise the likelihood of occurrence of events such as the 2015/16 Tasman Sea MHW. We have performed an extreme event attribution analysis to assess the role of climate change in modifying the likelihood of an event of this intensity and duration. The approach follows Lewis and Karoly[51], and King et al.[52], and involves the calculation of the Fraction of Attributable Risk (FAR) of an extreme event due to anthropogenic climate change using seven Coupled Model Intercomparison Project Phase 5 (CMIP5) global climate models (see Supplementary Table 2). The role of anthropogenic climate change was quantified using FAR values by comparing the distribution of MHWs in the historical simulations (climate model experiments including both natural and anthropogenic

forcing to reconstruct the 1850–2005 climate) and RCP8.5 simulations (climate model experiments projecting natural and anthropogenic climate variability into the twenty-first century) against the historicalNat simulations (identical to the historical simulations except excluding anthropogenic forcing, see Methods for more details). We performed two independent attribution analyses: one on the maximum intensity and one on the duration of the MHWs. We chose values of the second-largest maximum intensity and second-longest MHWs from the observed record of the SEAus region against which to make the attribution statement. Relative to an early 1881–1910 base period, which is required to test the importance of long-term anthropogenic climate change, the intensity of the event being attributed was 3.1 °C and the duration 446 days. These values are higher than those for the 2015/16 event based on the recent period (1982–2005) due to warming between the 1881–1910 and 1982–2005 periods ($+0.89$ °C). The choice of using the properties of the second-largest events as the critical values, as opposed to the largest event, reduces selection bias, provides a more conservative analysis and yields a larger sample size and thus greater confidence in FAR statements[12,51]. These FAR values therefore represent the risk due to anthropogenic climate change of an event longer or more intense than the one chosen, such as the 2015/16 event[52].

The FAR value for an event with maximum intensity of at least $+3.1$ °C was calculated to be 'very likely' (10th percentile of bootstrapped samples across estimates from various models and ensemble members, that is, 90% lower confidence bound based on Intergovernmental Panel on Climate Change categories[53], see Methods) at least 0.27 (best estimate, that is, median value: 0.65), for the historical runs from 1982 to 2005. This means an event of this intensity was very likely to be at least 1.4 times (best estimate: 2.9 times) as likely in the historical simulation period 1982–2005 compared with the historicalNat simulations over the period 1850–2005. Using the RCP8.5 simulations from 2006 to 2020, the FAR value was very likely to be at least 0.85 (best estimate: 0.91), indicating that this event increased to being very likely to be at least 6.8 times (best estimate: 11 times) as likely to be compared with the historicalNat simulations. From the distribution of FAR values we can say that it was very likely to be ($>90\%$) and virtually certain ($>99\%$) that anthropogenic climate change increased the likelihood of an event of this intensity in the 1982–2005 and 2006–2020 periods, respectively (Fig. 7c; thick black and red lines, respectively).

The FAR value for an event with a duration of at least 446 days was calculated to be very likely to be at least 0.992 (best estimate was 1 to three decimals of precision) for the historical runs over the period 1982–2005. This means an event of this duration was very likely to be at least 130 times as likely in 1982–2005 compared with the naturally forced simulations (best estimate: virtually impossible without anthropogenic forcing). Over the 2006–2020 period, the FAR value was very likely to be at least 0.997 (best estimate was one to three decimals of precision), indicating this this event increased to being very likely to be at least 330 times as likely compared with the naturally forced simulations (best estimate: virtually impossible without anthropogenic forcing). From the distribution of FAR values (Fig. 7d) it is very likely to be ($>90\%$) and virtually certain ($>99\%$) that anthropogenic climate change increased the likelihood of an event of this duration over the 1982–2005 and 2006–2020 periods, respectively (thick black and red lines, respectively).

We also examined how FAR values of the intensity and duration of the 2015/2016 Tasman Sea MHW have changed over time, including the projected change into the future. The distribution of FAR values from historical simulations indicate that, by the 1950–1980 period, it was already very likely to be

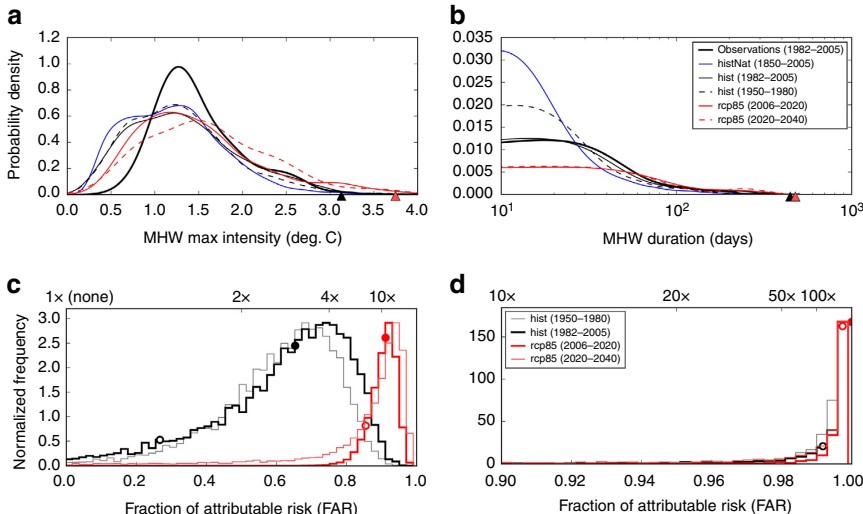

**Figure 7 | Attribution of the 2015/16 Tasman Sea marine heatwave event using global climate models.** The PDFs of the (**a**) maximum intensity and (**b**) duration of all MHWs detected from the observations (thick black line) and the ensemble of CMIP5 historical simulations over 1982–2005 (thin black line) and over 1950–1980 (dashed black line), historicalNat simulations (blue line), and RCP8.5 simulations over 2006–2020 (red line) and over 2020–2040 (dashed red line), using a baseline climatology of 1881–1910. The black and red triangles indicate the properties of the event being attributed and of the 2015/16 event, respectively. The distribution of FAR values for a MHW of (**c**) maximum intensity 3.1 °C or (**d**) duration of 446 days from the historical and RCP8.5 runs over four separate time periods. 1× (None) indicates no change in likelihood. The best estimate (median) and 10th percentile FAR values are indicated for the 1982–2005 and 2006–2020 periods by filled and open circles, respectively.

(>90%) that anthropogenic climate change increased the likelihood of an event of this intensity occurring, and likely to be (>66%) that it increased the likelihood of an event of this duration occurring (Fig. 7c,d, thin grey line). For future scenarios, the RCP8.5 simulations indicate that by the 2020–2040 period it will be likely to be (>66%) and very likely to be (>90%) that anthropogenic climate change will increase the likelihood of an event of this duration or intensity occurring, respectively (Fig. 7c,d, thin red line).

**Ecological impacts**. The observed ecological impacts of the 2015/16 Tasman Sea MHW appeared to be restricted to sessile, sedentary or cultured species in the shallow coastal near-shore environment and no observations of widespread mortalities of mobile species were reported from the Tasmanian coastline. In January 2016, the first Tasmanian outbreak of an oyster disease (Pacific Oyster Mortality Syndrome) was recorded and led to closed local hatcheries and decimated juvenile Pacific oyster (*Crassostrea gigas*) stocks. Tests on frozen oysters indicated that the virus was present off Tasmania since at least mid-December 2015 but was absent in March 2015 when previous testing occurred[54]. Previous outbreaks of this disease first appeared in New South Wales in 2010, following its first detection in France in 2008, and both have been linked to anomalously warm water[55].

Dead blacklip abalone (*Haliotis rubra*) were also observed in early March 2016 during research surveys in southeast Tasmania, at a magnitude of ~5% (IMAS, unpublished data), compared with zero mortality usually observed in surveys. Ongoing low-level mortalities of *H. rubra* were observed across most of the east and southeast Tasmanian coast until mid-April 2016. Although no mortalities were observed on the south or west coasts at any time during the MHW, abalone processors reported abalone in poor condition across all of southern Tasmania for the period December 2015 to March 2016, with above average mortality experienced during processing and live export.

Interestingly, abalone mortality occurred at temperatures during the peak intensity of the MHW that were more than 7 °C below the thermal maximum of 26.9 °C for this species[56], indicating there may be local adaptation by south-east Tasmanian abalone populations to the local cooler maximum summer temperatures. Investigations of metabolic activity of abalone under simulated harvest stress from New South Wales and southern Tasmania populations were consistent with a hypothesis of adaptation to local thermal regimes[57]. The temporal extent of the 2015/16 MHW (251 days) may have elevated background metabolic rates of abalone above normal for an extended period, reducing stored energy and consequently resilience to stress. The peak temperatures occurred at the end of the MHW, potentially creating an acute stress event for cooler adapted abalone populations, resulting in the observed mortality of wild abalone. A subsequent cut to the abalone quota for 2017 has been linked to the extra mortality as a result of this MHW.

In addition, the warm water temperatures in southeast Tasmania reduced performance in cultured Atlantic salmon that limited supply to seafood markets[58]. The warm temperatures were also associated with out-of-range observations of several fish species, including yellowtail kingfish, snapper, dusky morwong, mahi mahi, blue moki and moonlighter fish, which may have been carried southward by the anomalously strong East Australian Current Extension. Although these species have been occasionally found in Tasmanian waters[59,60], the number of species and sightings were higher than in recent years. Recreational fishers were particularly excited by the presence of kingfish and mahi mahi[61,62], indicating that some impacts are viewed as positive outcomes.

The impacts of this MHW can be compared and contrasted with other major events that have occurred recently. For example, the 2011 Ningaloo Niño event off Western Australia resulted in mortalities of up to 99% in the abalone (*Haliotis roei*) fishery and physically that MHW event seemed to have a similar pattern of temperature anomalies of up to +3 °C persisting over several months[63] as for the Tasman Sea MHW. The contrasting ecological impacts of these two MHWs on wild abalone populations reflect that the hardest hit *H. roei* abalone fishery was at the northern extent of its range in Western Australia with

temperatures up to 5 °C above average[64], whereas in Tasmania the southern extent of the abalone thermal and geographic range was affected. Even with local adaptation to cooler summer temperatures, there is likely to be a greater capacity of these southern populations to endure locally extreme thermal events if they are well below thermal maximum for that species.

## Discussion

Extreme events attract interest from scientists, resource users and the public at large—questioning the what, why and how of these events. At the time of an event, such as a MHW, there is interest in the size and duration, which can be reported in near-real time, based on satellite and other data sources. The 2015/16 MHW in the Tasman Sea was the longest (251 days) and most intense (+2.9 °C maximum anomaly) event on record in this region. This event was identifiable in daily remotely sensed SSTs, monthly gridded *in situ*-based SSTs and daily *in situ* near-shore sub-surface loggers. There is also keen interest in the why—that is, the cause of the event. This MHW coincided with anomalous southward flows and enhanced eddy kinetic energy corresponding to a strengthened southward extension of the East Australian Current. A temperature budget, in which we consider horizontal advection and air–sea heat flux as possible physical drivers, indicated (anomalous) southward advection to be the primary driver of the anomalous temperatures, consistent with a stronger southward extension of the EAC. Finally, there is interest in how climate change might influence or cause these events. An event attribution analysis using global climate models indicated that it was very likely to be that the occurrence of an event of this duration or intensity in the Tasman Sea region was ≥330 and ≥6.8 times as likely to be respectively due to the influences of anthropogenic climate change, compared with a naturally varying world. Impacts on marine ecosystems were varied and significant including an outbreak of Pacific Oyster Mortality Syndrome in Pacific oysters, mortality of blacklip abalone, poor performance of salmon aquaculture and intrusions by fish normally seen in warmer, more northerly waters. These multiple lines of evidence represent a comprehensive characterization of this extreme MHW event.

Examining ocean temperature allows characterization of MHW events and their impacts on coastal systems, but concurrent changes in other parameters, including salinity, nutrients and so on may moderate or exacerbate the effects of an extreme event. There was evidence of stress across the abalone, lobster, oysters and salmon industries during this event. All of the affected species (except salmon) tolerate much warmer temperatures elsewhere in their natural range than experienced at the peak intensity of this marine heatwave. In addition, the location of a MHW is important relative to the geographic distribution of the impacted species. Towards the upper end of a species' thermal tolerance range, a MHW can result in acute and catastrophic mortality, as in the Western Australia event in 2011 (ref. 11). In regions below a species' thermal tolerance, in the cooler part of its range, a MHW can result in chronic stress leading to some mortality, but generally results in reduced species performance as reported here. A chronic stress-causing MHW will exacerbate post-harvest issues in seafood industries, requiring greater attention to transport, holding and export processes. Importantly, this MHW has shown that we can expect impacts of climate change more broadly rather than just when it approaches species' thermal maxima.

We have applied a systematic approach to characterize regional MHW properties, causality and impacts taking account of multiple lines of evidence and application of a consistent definition. This approach can be applied in near-real time as these events evolve. This is in contrast to previous studies, which have separately examined events' physical drivers[3–7], ecological impacts[7–11] or the role of climate change[9–12]. In doing so we have continued the established use[5,6] of a temperature budget to diagnose physical drivers and pioneered the use of event attribution, which is used routinely in atmospheric science[51,52], to examine the role of anthropogenic climate change on the 2015/16 Tasman Sea MHW event. We note the importance of a remote-sensing observing system to track the evolution of marine heatwave events and long records of *in situ* nearshore measurements to monitor coastal impacts. This multi-disciplinary work has shown that MHWs can occur rapidly, have widespread impacts on wild fisheries and aquaculture industries, and on the broader marine ecosystem. It would be valuable for the approach taken here to become part of a real-time system that could provide alerts to the emergence and evolution of a marine heatwave, thus supporting the adaptive management of marine resources in these systems.

## Methods

**Observations of the ocean and atmosphere.** SST observations were obtained from two sources. We used the National Oceanic and Atmospheric Administration Optimum Interpolation Sea Surface Temperature (NOAA OI SST) V2 data set, which provides daily SSTs on a 0.25° grid[65]. These data are an interpolation of remotely sensed SSTs from the Advanced Very High Resolution Radiometer imager onto a regular grid. We obtained NOAA OI SST data over the period 1 January 1982 to 5 July 2016. We also used the HadISST data set, which provides monthly SSTs on a 1° grid[66]. These data are an interpolation of *in situ* observations onto a regular grid. We obtained HadISST data over the period January 1880 to May 2016. Time series of SST off southeast Australia were generated from both data sets by spatially averaging SST on each day or month over the region bounded by the longitudes [147°E, 155°E] and latitudes [45°S, 37°S], referred to as the SEAus region (Fig. 1a, black box). SST anomalies within this box were calculated by removing a climatology that was estimated by harmonic regression of the SST time series onto the long-term mean and the annual and semi-annual cycles.

Subsurface ocean temperature observations were obtained from a number of temperature loggers situated around Tasmania (Supplementary Table 1). The Maria Island Time Series, part of the Australian National Mooring Network run by the Integrated Marine Observing System (IMOS), provides Water Quality Monitor measurements of water temperature at 19 and 85 m depth at 15 min intervals. These data were obtained from IMOS (www.imos.org.au). Daily averages were calculated from the measurements to provide a daily time series from 31 July 2008 to 6 March 2017. *In situ* temperature measurements (Onset HOBO U22-001) from a further 12 fixed sites along the east coast of Tasmania were obtained from the Institute for Marine and Antarctic Studies (University of Tasmania) long-term inshore water temperature monitoring programme. These loggers sit ∼1 m off the sea floor in water depths ranging from 6 to 13 m and measure temperature at hourly or two-hourly sample rates. Time series of daily means were calculated as above.

The Maria Island Time Series also provided Acoustic Doppler current profiler measurements of zonal and meridional currents throughout the water column. We removed tidal components using a 39-point Doodson X0 filter[67], calculated daily and monthly means as with the temperature data and then averaged over depths between 2 and 74 m. This provided daily and monthly time series of depth-averaged meridional currents from 21 July 2011 to 13 May 2016.

Observed ocean surface circulation data were obtained from the IMOS OceanCurrent gridded sea-level anomaly product. This product includes an estimate of surface geostrophic velocities over the Australasian region optimally interpolated onto a 0.2° grid and we obtained daily data for the period 1 September 2011 to 31 May 2016. The geostrophic velocity anomalies were derived from sea-level anomalies and the mean surface velocity from 18 years of Ocean Forecasting Australia Model version 3 model output. More information can be found at http://oceancurrent.imos.org.au. Eddy kinetic energy was derived from velocities by first removing the seasonal climatology and then calculating, for each time point: Eddy kinetic energy $= 0.5(u^2 + v^2)$, where $u$ and $v$ are zonal and meridional velocities, respectively.

Atmospheric variables were obtained from the NCEP CFSv2 (refs 32,33). Air temperature and winds on the model Hybrid Level 1, which is a terrain-following pressure level near the Earth surface, were taken to represent surface variables. We obtained six-hourly data from the analysis step of the model, on a 0.2° grid, and calculated daily averages over the period 1 January 2012 to 31 May 2016. Anomalies were calculated by removing a climatology estimated by harmonic regression of the daily time series onto the annual and semi-annual cycles.

**MHW definition.** We used the Hobday *et al.*[2] definition to identify and quantify MHWs from daily temperature measurements[2]. In this definition, a MHW is

defined as a discrete prolonged anomalously warm water event. Specifically, discrete implies the MHW is an identifiable event with clear start and end dates, prolonged means it has a duration of at least five days and anomalously warm means the water temperature is warm relative to a baseline climatology. Quantitatively, marine heatwaves were identified as periods of time when temperatures were above the seasonally varying 90th percentile (the threshold) for at least five consecutive days; two successive events with a break of 2 days or less between were considered a single continuous event. The seasonally varying mean (climatology) and 90th percentile threshold were calculated for each day of the year using daily temperature values across all years and within an 11-day window centred on the day, and were then smoothed using a 31-day moving window. The period used to define the climatology was 1982–2005 for the NOAA OI SST data; for the *in situ* temperature logger data the climatology period was set to the total available record at each station.

MHWs have a set of metrics used to describe their properties[2]. Here we considered the following metrics: duration (the time between the start and end dates), mean intensity and maximum intensity (the average and maximum temperature anomaly over the duration of the event) and cumulative intensity (the integrated temperature anomaly over the duration of the event). Anomalies were measured relative to the seasonal climatology. We used a software implementation of this definition freely available in the Python programming language (http://github.com/ecjoliver/marineHeatWaves).

**Upper ocean temperature budget.** A temperature budget was used to compare the relative roles of horizontal advection and air–sea heat flux to the upper ocean warming during the event. The temperature tendency equation was volume-averaged over a depth $h$ and an area $A$ defined over the SEAus domain (Fig. 1a, box), yielding the following expression:

$$\underbrace{\frac{\partial \langle T \rangle}{\partial t}}_{\text{RATE}_V} = \underbrace{-\langle \boldsymbol{u}_H \cdot \boldsymbol{\nabla}_H T \rangle}_{\text{ADV}_H} + \underbrace{\frac{1}{A} \int^A \frac{Q}{h} \mathrm{d}A}_{Q_V} + \underbrace{\langle \boldsymbol{\nabla}_H \cdot (\kappa_H \boldsymbol{\nabla}_H T) \rangle - \langle w \frac{\partial T}{\partial z} \rangle - \frac{1}{A} \int^A \left( \kappa_V \frac{\partial T}{\partial z} \right)_{-h} \mathrm{d}A}_{\text{Residual}}$$

(1)

and $\langle \cdot \rangle = \frac{1}{hA} \int^A \int_{-h}^0 \cdot \, \mathrm{d}z \mathrm{d}A$. The temperature is $T$, $\boldsymbol{u}_H$ are the horizontal (zonal and meridional) currents, $w$ is the vertical current, $\boldsymbol{\nabla}_H$ is the horizontal gradient operator, $\kappa_H$ is the horizontal diffusivity, $Q$ is the total air–sea heat flux (the net contribution due to sensible and latent heat flux, as well as short- and long-wave radiation), and $\kappa_V$ is the vertical diffusivity. The temperature budget is integrated over a depth $h$ of 100 m chosen to capture the bulk of the upper ocean warming signal.

Estimates of the three-dimensional ocean state were obtained from the Australian Bureau of Meteorology's Bluelink OceanMAPS analysis version 2.2.1 (ref. 31) (http://wp.csiro.au/bluelink), which is an operational continuation of the Bluelink ReANalysis system[68]. Daily sea level, temperature, salinity, and zonal and meridional velocities to 1,000 m depth on a $0.1° \times 0.1°$ grid were obtained for the period 1 January 2012 to 31 March 2016. OceanMAPS is forced by ACCESS-G surface fluxes and, as this product was not available for comparison, we obtained the total surface heat flux at the ocean surface daily from the NCEP CFSv2 (refs 32,33) and monthly from the GODAS[34,35] over the same time period.

Following Benthuysen *et al.*[5], RATE$_V$ is the time-rate of change in volume-averaged temperature, ADV$_H$ is the time-rate of change in volume-averaged temperature due to horizontal advection and $Q_V$ is the time-rate of change in volume-averaged temperature due to air–sea heat flux. The Residual term has contributions from lateral diffusion, vertical temperature advection and entrainment. This term was not calculated as the required variables were not available as output, but the temperature budget results shows that the Residual did not provide a dominant contribution to RATE$_V$. The contributions to the change in volume-averaged temperature $\langle T \rangle$ were determined by integrating the terms RATE$_V$, ADV$_H$ and $Q_V$ in time from 1 September of each year (denoted $T_V$, $T_H$ and $T_Q$, respectively), which is shown for 2015/16 in Fig. 6 and Supplementary Fig. 6.

We calculated the climatology of the temperature budget by averaging $T_V$, $T_H$ and $T_Q$ across all years (before the 2015/16 event, that is, 2012/13, 2013/14 and 2014/15) and using an 11-day window centred on each day-of-year. The climatology only used data back to 2012, as that is the start of available OceanMAPS data. Temperature budget anomalies for 2015/16 were calculated as the difference between $T_V$, $T_H$ and $T_Q$ for that year and the climatology.

**Climate change and event attribution.** Surface ocean temperatures were analysed from a set of CMIP5 global climate models[69]. We extracted daily SSTs from the historical (1850–2005), historicalNat (1850–2005) and RCP8.5 (2006–2100) simulations from the six models listed in Supplementary Table 2. The historical simulations include both natural forcing (volcanoes and solar) and anthropogenic forcing (greenhouse gases, aerosols and ozone) to realistically simulate the climate over the 1850–2005 period; the historicalNat simulations include only natural forcing to simulate an alternate climate over 1850–2005, which lacks anthropogenic forcing. By comparing the historical and historicalNat simulations, we can quantify the impacts of anthropogenic climate change. The RCP8.5 (Representative Concentration Pathway high emissions scenario) simulation is a future projected

climate experiment under a scenario of high emissions of greenhouse gases into the twenty-first century, continuing from the end of the historical simulations. We used the RCP8.5 future projections, as they best represented the observed emissions since 2006 (ref. 70). For each model simulation the mean daily SST over the SEAus region was calculated. The need for daily data, to implement the MHW definition, restricted our selection of models to those listed in Supplementary Table 2, as SSTs from most models were only available monthly, particularly for the historicalNat experiment. The observations used to validate the historical simulations were the NOAA OI SST data over the period 1 January 1982 to 5 July 2016, averaged over the SEAus region.

For the attribution of extreme events, the observed and historical model distributions of the parameter being attributed (for example, MHW intensity or duration) must be similar for the models to realistically represent plausible climate variability. Therefore, each model ensemble member was bias-corrected to best reflect the observed non-seasonal SST variability. The importance of the non-seasonal variability is its contribution to the properties of MHWs (duration, intensity and so on). We decomposed the observed and model time series into a sum of mean, linear trend, seasonal and non-seasonal components:

$$T_t = a + bt + T_t^S + T_t'$$

(2)

where $T_t$ is SST at some time $t$, $a$ and $b$ are the linear intercept and slope parameters, $T_t^S$ is the seasonal cycle and $T_t'$ is the residual, non-seasonal component. The intercept and slope parameters were estimated by linear regression and the seasonal cycle by harmonic regression. The linear and seasonal components were subtracted from $T_t$ to isolate $T_t'$. The s.d. of $T_t'$ was denoted $\sigma$. For each model ensemble member, the ratio of $\sigma$ from the historical run to $\sigma$ from the observations was calculated, restricted to the shared period of 1982–2005 and an ensemble mean of this ratio was calculated for each model. This ratio was used to bias-correct the variance of $T_t'$ for each model and this bias-corrected $T_t'$ was then added back to the mean, linear trend and seasonal cycle components to yield the bias-corrected total SST time series. The computed values of the $\sigma$ ratio can be found in Supplementary Table 2.

MHWs were detected from the observations and the historical, historicalNat and RCP8.5 runs. We used the 1881–1910 period to define the climatology, a period that was relatively well-observed (Supplementary Fig. 8) and suitably early so as to provide a benchmark against which any signature of anthropogenic warming over the period can emerge and the historical run climatology was used for the RCP8.5 runs[51]. For the observations this choice was problematic, as no satellite SST observations exist before 1982. We therefore estimated the observed climatology by first calculating it over the 1982–2005 period and then correcting for the mean warming since the 1881–1910 period ($+0.89 °C$), which was calculated from the HadISST observational data set. This method assumes no change in the variability of SST between these two time periods, which was confirmed by examining the 30-year running s.d. of SST from HadISST (Supplementary Fig. 9). The s.d. of SST shows some multi-decadal variability but no significant secular trend or difference between the two time periods of interest here. As the 2015/16 event is now measured relative to a cooler base period, the metrics used to describe it have changed accordingly to an intensity of 3.7 °C and a duration of 477 days. This facilitated a direct comparison of the model and observations and an examination of the impact of century-scale climate change on MHWs.

Probability density function (PDFs) of MHW duration and maximum intensity were calculated from the properties of the detected events from the observations and each of the climate model experiments described above (each ensemble member was weighed by the inverse of the number of ensembles for that particular model, thereby weighting equally across models). A kernel density estimate using a Gaussian kernel with bandwidth determined by Scott's rule[71] was used to estimate the PDFs. The PDFs were generated from subsets of the detected MHWs to isolate particular time periods (that is, 1950–1980 and 1982–2005 from the historical runs and 2006–2020 and 2020–2040 from the RCP8.5 runs; Fig. 7a,b).

A FAR analysis[72] was performed using the historicalNat, historical and RCP8.5 runs. First, we chose the second-largest duration and maximum intensity from the record of observed MHWs, which were 446 days (which occurred over 30 July 2011–17 October 2012) and 3.1 °C (which occurred on 3 February 2001) respectively, on which to make the attribution statements. The choice of using the second-largest events as the critical values, as opposed to the largest event, reduces selection bias, provides a more conservative analysis, and yields a larger sample size and thus greater confidence in FAR statements[12,51]. It also allows us to answer whether anthropogenic climate change increased the likelihood of events longer or more intense than these events, such as the 2015/16 MHW[52]. Then, separately for duration and maximum intensity, we defined the FAR for an event with this critical value

$$\text{FAR} = 1 - \frac{P_{\text{histNat}}}{P_{\text{hist}}}$$

(3)

where $P_{\text{histNat}}$ is the probability of an event at least as long (or intense) as the critical value. It is noteworthy that $P_{\text{histNat}}$ was calculated over the full 1850–2005 period, to accurately represent the likelihood of this event due solely to natural variability, but $P_{\text{hist}}$ was calculated over subsets (1950–1980 and 1982–2005) to isolate how the FAR changed with increasing anthropogenic influence on the climate (for example, ref. 73). In addition, we can replace $P_{\text{hist}}$ with $P_{\text{RCP8.5}}$ to

examine how the FAR may change in a projected future climate using the RCP8.5 runs (2006–2020 and 2020–2040). The ensemble members used to calculate the FAR values were pooled together and bootstrapped (with replacement, the likelihood of selecting an ensemble member weighted by the inverse of the number of ensembles for that particular model as above) 10,000 times with a sample size of 14 (half of the total number of historical ensemble members: 28). This provided 10,000 FAR values and from these the confidence bounds on the FAR results were calculated[51,52]. The lower confidence bounds and the proportion of positive FAR values were reported using Intergovernmental Panel on Climate Change AR5 terminology[53] against the likelihood of an increase of risk due to anthropogenic climate change (that is, very likely to be as >90%, virtually certain as >99% and so on). It is noteworthy that as the baseline period was chosen to be 1881–1910, all statements about the influence of anthropogenic climate change implicitly include the qualifier: since the 1881–1910 baseline period.

**Reference periods.** It is noteworthy that a number of different reference periods were used throughout this study and these are clarified here. The observational description of the event from the NOAA OI SST data is relative to the 1982–2005 period (from the start of the data to the end of the CMIP5 historical period, to enable comparison). The nearshore temperature logger data anomalies are each relative to the period corresponding to each logger's full time span (see Supplementary Table 1). The CFSv2 anomalies are relative to the full period of data obtained (1 January 2012 to 31 May 2016). In the climate change attribution analysis, all results are presented relative to the 1881–1910 period (an early period, before significant anthropogenic climate change, that is also well-observed in HadISST).

**Data availability.** We have made use of publicly available data only; no new data were generated as a result of this study. NOAA High Resolution SST and GODAS data provided by the NOAA/OAR/ESRL PSD (Boulder, Colorado, USA) from their web site (www.esrl.noaa.gov/psd/). Hadley Centre SST data were provided by the Met Office (UK) from their website (www.metoffice.gov.uk/hadobs). Maria Island Time Series and OceanCurrent data were sourced from IMOS—IMOS is a national collaborative research infrastructure, supported by the Australian Government. NOAA High Resolution SST data provided by the NOAA/OAR/ESRL PSD (Boulder, Colorado, USA) from their web site (www.esrl.noaa.gov/psd/). The code used to analyse these data and generate the results presented in this study can be obtained from https://github.com/ecjoliver/TasmanSeaMHW_201516.

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

## Acknowledgements
E.C.J.O. was supported by Australian Research Council (ARC) grant number CE110001028. S.E.P.-K. was supported by ARC grant number DE140100952. This study makes a contribution to the ARC Centre of Excellence for Climate System Science (ARCCCS; Grant number CE110001028), the National Environmental Science Programme (NESP) Earth Systems and Climate Change (ESCC) Hub Project 2.3 (Grant number B0024391) and the International Commission on Climate of IAMAS/IUGG. Finally, we acknowledge and thank the reviewers (Fernando Lima, Dáithí Stone and one anonymous): your comments led to significant improvements to this manuscript.

## Author contributions
E.C.J.O. led and coordinated the various components of the study. E.C.J.O. and J.A.B. designed and performed the physical drivers analysis. N.J.H. synthesised the ENSO knowledge. E.C.J.O. and S.E.P.-K. designed and performed the climate change event attribution analysis. A.J.H. and C.N.M. wrote the ecological impacts section. All authors discussed the results, aided in their interpretation and contributed in writing the paper.

## Additional information

**Competing interests:** The authors declare no competing financial interests.

