## [Peer Review File · Nature Communications]

Reviewers' comments:

Reviewer #1 (Remarks to the Author):

This manuscript by Oliver et al. was a pleasure to read. These authors made a quite comprehensive study of the marine heatwave that occurred in the Tasmanian Sea in 2015-2016. Each one of the approaches they took is not innovative per se, but combining all those perspectives in a single study is impressive and will certainly become a reference to which other papers will be compared in the future.

The English is good and the tone of the paper is also good. It has a lot of detail which makes it quite long but that is necessary because the strength of this paper would be lost if authors would be forced to cut pieces of the study. And each one of those pieces needs to be properly explained (to assure, for example, the reproducibility of the study), so I think there's no way to make the paper shorter. I am also happy with the statistical analysis that were done.

Honestly, my job as reviewer was quite facilitated by the MS, and I do not have a lot to correct or criticise. I made a few comments in the PDF and in the word document that I am attaching to this review. Basically, there are some places with small English errors and other places that are missing references. Please feel free to share my comments with the authors along with my identity. I am a fan of transparency in the peer-review process.

Apart from what is written in the PDF (and in the word document), I think authors should make very clear which reference periods they use for calculating the anomalies. That varies through the paper accordingly to the dataset that is being analysed and obviously, it is not possible to use a single reference period. The reader, however, soon becomes lost and should be helped with more clear statements of the periods and also with explanation for why were those period chosen over any other periods. The reference periods have a strong influence on the resulting trends and so the rationale behind each one must be stated.

I also did not understand why there was the need to smooth out some of the data. See the PDF for more details.

In summary, I enjoyed the paper and it only needs minor corrections in order to be accepted for publication (which I hope will happen).

Fernando P. Lima

Reviewer #2 (Remarks to the Author):

This paper presents report describing various facets of the 2015/2016 marine heatwave in the Tasman Sea. I think it is worthy of publication in Nature Communications, but first needs some revisions. These revisions may not be too substantial in terms of implementation, but I think they are required.

First, the abstract does not present the paper accurately, I think. From the abstract it seemed like this was a single study, with the ultimate goal of understanding the human influence on the Tasman Sea marine heatwave. But it is rather three studies which do not inform each other: one ("observations") describing the physical nature of the heatwave that occurred; one ("attribution") diagnosing the human influence; and one ("impacts") describing suspected impacts. The "observations" subpaper does not inform the "attribution" subpaper, e.g. by forming criteria for evaluation of whether the climate models are fit-for-purpose (because this heatwave was a case of a moving boundary, biases in that boundary dictate whether the process is represented in the

models or not). Neither the "attribution" nor "impacts" subpapers inform any of the other subpapers, and the "impacts" subpaper is not informed by the other two either. Is this a badly designed paper then? It seems like it by the impression given in the abstract (and to some degree the introduction), which presents the paper as a coherent analysis. But I think that is a misrepresentation. This is really instead a (relatively) fast "reporting on/describing various aspects of the heatwave" paper, in the same vein as van der Wiel et alii (2016, 10.5194/hess-2016-488). Therefore the unifying theme is the physical event itself, consisting of a collation of the standard disciplinary reports on the physical and impacts side of an extreme weather event, and an expansion to include a climate attribution component. This is thus expanding the concept of operational reporting. There are potentially pros and cons to this, but overall I think it is a novel concept/product that both warrants dissemination to a multi-disciplinary audience (as in Nature Communications) and would benefit from resulting discussions. This message finally comes through in the final conclusion section, but by that time I had already critically viewed the paper from the wrong angle (and thus maybe had been overly pedantic in my review below).

Second, there are a number of clarifications that need to be made on aspects of the various analyses. I have highlighted these in the specific comments below.

Third, a number of (to me) odd decisions were made in the approach in the "attribution" subpaper. I have highlighted these in the specific comments below. These particularly probably affect the estimates of the confidence range in the quantitative results, some decreasing and some increasing the range, relative to other more obvious approaches. Either some more obvious decisions should be used or a discussion of the implications of these decisions should be included.

Specific comments:

line 22

Some context is needed here. One of the strongest El Nino events ever recorded was taking place at that time, and from the point of view of marine life in the eastern tropical Pacific that was a pretty "long and intense" event.

lines 28-29

Some indication of the range of uncertainty on these numbers is needed.

line 29

"and event" -> "and an event"

line 29

"Future projections" -> "Projections of the future"

lines 38-39

Is "either the longest or most intense ever recorded" since 1982 as well? This is ambiguous.

lines 51-52

"This event impacted... detrimental stressors on coastal fishery and aquaculture industries including abalone, Pacific oysters and Atlantic salmon". The examples sound like the species being fished rather than stressing the fishery.

lines 51-56

Do you have evidence, e.g. references, to back up these assertions?

line 106

So there was a bigger one in 1982?

lines 106-108

How much local data is informing the circa 1900 HadISST estimates? Is it interpolation/extrapolation? There is less month-to-month variability, which makes me wonder if it is highly interpolated.

lines 136-137

The "strong southward surface flow" is not clear in Figure 4. The caption indicates that colours indicate actual current speeds (but it looks to me like they indicate current speed anomalies), but there is no directional information. The arrows represent anomalies, so we cannot interpret them in terms of actual currents without knowing the reference current map.

line 147

Figure 4 is not much help here, because it shows anomalies, not actual currents.

lines 160-161

See line 147 comment.

lines 190-191

I am a bit confused as to what mechanisms are included in "air-sea heat flux" here. Presumably sensible and latent heat mechanisms and longwave radiation are included. But is shortwave radiation? That has bypassed the "air" entirely.

lines 235-237

Why are you mentioning this, seeing as you are using the largest event, not second-largest, to set your threshold (as stated in lines 231-233 and 238-239)?

lines 231-233

I am not sure why you are using this reference from the distant past for estimating the threshold for the attribution calculations. If one is using an absolute (rather than return-period-based) threshold, then the need for a reference period is to remove model-observation bias. As far as the model simulations are concerned, it does not matter which period you choose: there is, by definition (ignoring drift), no bias between the historical and natural historical scenarios. But for the observations it does matter: the quality of the record is much poorer during some periods. It seems to me that you have chosen a period with nearly the poorest quality possible within the 20th century.

lines 242-243

What does this confidence interval represent? Is it the spread across estimates for various models?

line 243

The rounding error here is misrepresenting the results. $FAR = -0.17$ equals a Risk Ratio of 0.85, so following the number of significant digits presented in the FAR value, the 90% confidence on the RR value is consistent with a decrease, i.e. an $RR < 1$.

lines 244-245

What sort of bias might be introduced by using different periods for the two scenarios?

lines 287-288

So was this a just a coincidence then?

line 301

Where does the "99%" value come from?

line 308

How is that deduced?

line 316

Do you have evidence to support "there is also a keen interest"?

lines 323

It took me a little while to match these values to the calculations in the text, because originally they were presented to one significant digit only, whereas here you introduce more precision.

lines 430 and 433

Do shortwave radiation fit in "air-sea heat flux". It is not a flux of heat between the two, but then I do not see where else it would go.

line 471

I think you mean "particularly for the historical experiment", because that has fewer simulations in your table.

lines 489-490

If I understood correctly, you are using an absolute threshold, which also means that the frequency of exceedence will be affected by relative bias in the mean ("a"), trend ("bt"), and annual cycle ("T_t^S") components. Did you account for any of these as well? It does not sound like it here.

lines 493-506

As above, I am not clear on why you needed the extra steps, which amongst other things introduce sampling variance and hence broaden the uncertainty on your results. Why not just use the well-observed 1982-2005 period and, for all scenarios, use the data from the historical 1982-2005 data for the calibration?

lines 507-509

This could introduce a bit of a problem if models respond differently, because you have inconsistently different sample sizes across scenarios. For instance, the estimate for the 1982-2005 period are only half-informed by the CNRM-CM5 model (nearly no historical simulations, five historicalNat simulations) but are well informed in the estimate for the 2006-2020 period. Similarly, for 1982-2005 you are essentially estimating historicalNat in part from CanESM2 and CNRM-CM5, but not for historical. If you are calling these "PDFs", then this sampling represents an unobvious distribution on your prior. There are a number of possible "uniform-like" priors, most of which would involve weighing models by the inverse of their sample size for a given scenario. In any case, this choice of prior needs to be justified.

lines 515-517

This agrees with lines 235-237, but contradicts the lines immediately preceding and following 235-237.

lines 532-533

Which means your estimated uncertainty range are probably about $\sqrt{2}$ larger than the uncertainty around the estimates value?

lines 536-537

Um, this is manipulating results. First, if this occurs less than 5% of the time it will not affect your estimate of the 90% confidence interval. Second, if that possibility is within the resolution of your frequency estimates, then that is the result. If we know that a 2015/2016-like event occurred in a historicalNat-like boundary condition in the real world, then we can state that a lower bound of 0/N frequency is impossible, but we cannot automatically infer than an actual frequency of less than 1/N is also impossible.

lines 540-542

So actually your historicalNat simulations are not a sample of a natural climate, but rather an extended sample of a 1910-1940-like climate with added variance from the extra bias correction steps required?

lines 728-729

For NOAA OI or HadISST?

Figures 3, 4, and 5

Anomalies from what reference period?

line 750

The current speed (colours) and the current speed anomalies (arrow lengths) look rather similar. Are the colours actually showing anomalies too?

line 770

These triangles are very hard to see.

Supp line 47

Add quotation marks to "Complete".

Sincerely,

Daithi Stone

Reviewer #3 (Remarks to the Author):

Review of "The unprecedented 2015/16 Tasman Sea marine heatwave" by Oliver et al.

This manuscript discusses an extreme temperature anomaly during 2015-2016 in the Tasman Sea. The paper, as the authors noted is a systematic documentation, which discusses the evolution, causal factors, impacts, and future reoccurrence of the extreme event. The major conclusion of this paper is that the southward advection of warm water was the dominant driver of the extreme event and the likelihoods of this kind of event will increase due to anthropogenic influence. The authors claim that this paper is "an advancement" of previous studies, which focused on different aspects (physical drivers, ecological impacts, and the role of anthropogenic forcing) of similar extreme events occurred in other parts of the world's ocean. I have reservations about this statement, because the scientific methods in this paper are mostly, if not all, based upon these previous studies and piecing together different aspects of a problem doesn't necessarily mean scientific advancement. Moreover, trying to cover every aspect of the problem in a short contribution might instead dilute the core science. With that said, this is a timely report on an increasingly important issue, which should be of interest to a diverse readership. I have some comments below to help the improvement of this manuscript.

Major comments:

1. Perhaps the most exciting scientific aspect of this paper is the investigation of the physical drivers of the extreme event. Looking at Figure 3 to Figure 6, I generally agree with the authors that ocean advection should be the primary driver. However, the key statement in the abstract "the warming was dominated by anomalous net southward advection linked to the East Australian Current" was not well supported. In Figure 6, TH is the total horizontal advective flux that contains advective fluxes from both directions and doesn't directly indicate the role of southward advection of the western boundary current. Figure 4 shows some southward surface geostrophic current anomalies, but doesn't directly show the depth-averaged (0-100m) heat divergence anomaly,

which should provide direct evidence to support the statement of southward advection. My suggestions would be to decompose the total advection term in Figure 6 to quantify the contributions in both directions, and/or to plot depth-averaged heat divergence anomaly.

2. The other important discussion of this paper is on the future reoccurrence of extreme temperature anomaly like 2015/2016. This was discussed from page 10-14. However, I found the discussions confusing in that the numbers in the text were not obviously supported by the figures. For example, in the key conclusion “an extreme warming event of this duration was 8.5 times as likely to occur due to anthropogenic climate change; and event of this intensity was 5.9 times as likely”, the numbers 8.5 and 5.9 were not discussed/explained in the text. I have listed some other similar issues below. This part would be much improved if the authors can be explicit about the derivation of the numbers.

Other comments/corrections:

1. I would always be cautious about using the word “unprecedented”. The fact that the magnitude of the anomaly was the largest on record doesn't mean it is unprecedented.

2. Page 6, line 107: “top 10 greatest” was not obvious from the Figure. Marking the top 10 months in the figure would help.

3. Page 6, lines 111-125: Any specific reason why the emphasis was placed on Maria Island mooring?

4. Page 11, lines 240-243: “The best estimate (i.e., median) FAR value ... to be 0.58, which based on Intergovernmental Panel on Climate Change (IPCC) ... is ‘very likely’ ... to be between -0.17 and 0.84 ...”. This sentence reads awkward. 0.58 is very likely to be between -0.17 and 0.84?

5. Page 11, line 242-244: The numbers -0.17, 0.84, 1, 6, and 2 are not clearly supported by the figures.

6. Page 11, line 244-245: “...historical simulation period 1982-2005 compared to the historicalNat simulations over the period 1850-2005”. Are the two periods directly comparable?

7. Page 12: Again, numbers were not clearly derived or marked in the figures.

8. Page 20, line 453 and Figure 6: Why only use 3 years to construct the climatology?

Response to Reviewers

We would like to thank all reviewers for their careful reading and constructive comments on this manuscript. We believe it is now much stronger after addressing their concerns. The reviewer's comments are listed point-by-point below with the reviewer's original comments in italics and our responses in bulleted roman text.

Response to Reviewer #1

This manuscript by Oliver et al. was a pleasure to read. These authors made a quite comprehensive study of the marine heatwave that occurred in the Tasmanian Sea in 2015-2016. Each one of the approaches they took is not innovative per se, but combining all those perspectives in a single study is impressive and will certainly become a reference to which other papers will be compared in the future.

The English is good and the tone of the paper is also good. It has a lot of detail which makes it quite long but that is necessary because the strength of this paper would be lost if authors would be forced to cut pieces of the study. And each one of those pieces needs to be properly explained (to assure, for example, the reproducibility of the study), so I think there's no way to make the paper shorter. I am also happy with the statistical analysis that were done.

Honestly, my job as reviewer was quite facilitated by the MS, and I do not have a lot to correct or criticise. I made a few comments in the PDF and in the word document that I am attaching to this review. Basically, there are some places with small English errors and other places that are missing references. Please feel free to share my comments with the authors along with my identity. I am a fan of transparency in the peer-review process.

- Done.
- The comments provided in the PDF and Word document have been implemented in the revised manuscript. Responses to the more detailed comments relating to reference periods and data smoothing can be found below.

Apart from what is written in the PDF (and in the word document), I think authors should make very clear which reference periods they use for calculating the anomalies. That varies through the paper accordingly to the dataset that is being analysed and obviously, it is not possible to use a single reference period. The reader, however, soon becomes lost and should be helped with more clear statements of the periods and also with explanation for why were those period chosen over any other periods. The reference periods have a strong influence on the resulting trends and so the rationale behind each one must be stated.

- Done.
- The following section has been added to the end of the Methods:
“**Reference periods** Note that a number of different reference periods were used throughout this study and these are clarified here. The observational description of the event from the NOAA OI SST data is relative to the 1982-2005 period (from the start of the data to the end of the CMIP5 historical period, to enable comparison). The nearshore temperature logger data anomalies are each relative to the period corresponding to each logger's full time span (see Supplementary Table 1). The CFSv2 anomalies are relative to the full period of data obtained (1 Jan 2012 to 31 May 2016). In the climate change attribution analysis all results are presented relative to the 1881-1910 period (an early period, before significant anthropogenic climate change, that is also well-observed in HadISST).”

- In addition, the reference period has now been explicitly stated in Figure captions where this information was previously lacking (e.g. Figs. 3 & 5).

I also did not understand why there was the need to smooth out some of the data. See the PDF for more details.

- The only smoothing undertaken was on the climatology, used to define marine heatwaves. This smoothing was implemented according to the recommendations of Hobday et al. (2016; see point #3 in “Recommendation and conclusion” of that paper). The smoothing was required in order to obtain a smoothly varying climatology, from highly variable daily data, such that the marine heatwave detection algorithm would not lead to erroneous detection of events.

Hobday AJ, LV Alexander, SE Perkins, DA Smale, SC Straub, ECJ Oliver, J Benthuisen, MT Burrows, MG Donat, M Feng, NJ Holbrook, PJ Moore, HA Scannell, A Sen Gupta and T Wernberg, 2016: A hierarchical approach to defining marine heatwaves. *Progress in Oceanography*, **141**, 227-238, doi:10.1016/j.pocean.2015.12.014.

In summary, I enjoyed the paper and it only needs minor corrections in order to be accepted for publication (which I hope will happen).

Fernando P. Lima

Response to Reviewer #2

This paper presents report describing various facets of the 2015/2016 marine heatwave in the Tasman Sea. I think it is worthy of publication in Nature Communications, but first needs some revisions. These revisions may not be too substantial in terms of implementation, but I think they are required.

First, the abstract does not present the paper accurately, I think. From the abstract it seemed like this was a single study, with the ultimate goal of understanding the human influence on the Tasman Sea marine heatwave. But it is rather three studies which do not inform each other: one ("observations") describing the physical nature of the heatwave that occurred; one ("attribution") diagnosing the human influence; and one ("impacts") describing suspected impacts. The "observations" subpaper does not inform the "attribution" subpaper, e.g. by forming criteria for evaluation of whether the climate models are fit-for-purpose (because this heatwave was a case of a moving boundary, biases in that boundary dictate whether the process is represented in the models or not). Neither the "attribution" nor "impacts" subpapers inform any of the other subpapers, and the "impacts" subpaper is not informed by the other two either. Is this a badly designed paper then? It seems like it by the impression given in the abstract (and to some degree the introduction), which presents the paper as a coherent analysis. But I think that is a misrepresentation. This is really instead a (relatively) fast "reporting on/describing various aspects of the heatwave" paper, in the same vein as van der Wiel et alii (2016, 10.5194/hess-2016-488). Therefore the unifying theme is the physical event itself, consisting of a collation of the standard disciplinary reports on the physical and impacts side of an extreme weather event, and an expansion to include a climate attribution component. The paper is thus expanding the concept of operational reporting. There are potentially pros and cons to this, but overall I think it is a novel concept/product that both warrants dissemination to a multi-disciplinary audience (as in Nature Communications) and would benefit from resulting discussions. This message finally comes through in the final conclusion section, but by that time I had already critically viewed the paper from the wrong angle (and thus maybe had been overly pedantic in my review below).

- Done.
- We acknowledge and, on reflection, agree with the reviewer's comment. We have endeavoured to shift the tone of the abstract and introduction to better reflect the nature of the work: i.e. that it combines the findings from analyses of several factors associated with this extreme marine heatwave event.
- We have added the following line to the abstract: "We report on several inter-related aspects of this event: observed characteristics, physical drivers, ecological impacts, and the role of climate change."
- The third paragraph of the introduction has been revised as: "This paper discusses the 2015/16 Tasman Sea marine heatwave from observations and ocean models, diagnoses its physical drivers and the role of anthropogenic climate change, and describes the ecological impacts that occurred. By collating the analyses together we can integrate the inter-related consequences of this extreme event from the physical drivers and climate change, and their impacts on marine ecosystems. Thus we illustrate how to characterise marine heat waves regionally, where there is a growing need for a clear and timely analyses for such events."
- In the last paragraph of the discussion, we have removed the original statement that the study represents an "advancement over previous studies". We have now revised this statement to read "This is in contrast to previous studies which have separately examined events' physical drivers³⁻⁷, ecological impacts⁷⁻¹¹ or the role of climate change⁹⁻¹²".
- [See also Reviewer #3, Comment #1]

Second, there are a number of clarifications that need to be made on aspects of the various analyses. I have highlighted these in the specific comments below.

Third, a number of (to me) odd decisions were made in the approach in the "attribution" subpaper. I have highlighted these in the specific comments below. These particularly probably affect the estimates of the confidence range in the quantitative results, some decreasing and some increasing the range, relative to other more obvious approaches. Either some more obvious decisions should be used or a discussion of the implications of these decisions should be included.

Specific comments:

line 22

Some context is needed here. One of the strongest El Nino events ever recorded was taking place at that time, and from the point of view of marine life in the eastern tropical Pacific that was a pretty "long and intense" event.

- Done.
- Re-phrasing to emphasize that it was locally the longest and most intense event.

lines 28-29

Some indication of the range of uncertainty on these numbers is needed.

- Done.
- These statements have been presented as the 90% lower confidence bound, rather than the best estimate (median) result.

line 29

"and event" -> "and an event"

- Done.

line 29

"Future projections" -> "Projections of the future"

- Done.

lines 38-39

Is "either the longest or most intense ever recorded" since 1982 as well? This is ambiguous.

- Done.
- Re-phrased to remove ambiguity.

lines 51-52

"This event impacted... detrimental stressors on coastal fishery and aquaculture industries including abalone, Pacific oysters and Atlantic salmon". The examples sound like the species being fished rather than stressing the fishery.

- Done.
- Re-phrased to remove ambiguity.

lines 51-56

Do you have evidence, e.g. references, to back up these assertions?

- Done.
- In the ‘Ecological impacts’ sub-section under results we previously noted the appropriate references for out-of-range species sightings, coastal fisheries stressors, and impacts on aquaculture. We have also now added a reference to redmap.org.au, a range extension sighting database where out-of-range species can be noted and uploaded by the public.

line 106

So there was a bigger one in 1982?

- Done.
- It was the largest on record, and records began in 1982. Removed “since 1982” from the text to make the statement clearer.

lines 106-108

How much local data is informing the circa 1900 HadISST estimates? Is it interpolation / extrapolation? There is less month-to-month variability, which makes me wonder if it is highly interpolated.

- The HadISST data is interpolated and therefore provides valid “data” in the presence of low (or no) observational density. An estimate of the presence of absence of underlying data can be obtained from the HadSST3 dataset which consists of monthly spatial means over 5-degree grid cells, based on the same *in situ* observations (Rayner et al. 2003). Averaging HadSST3 over the SEAus region provides a monthly SST time series with missing values when observations are completely lacking. Since 1960 there have been no missing months; and all other decades except the 1850s, 1860s and the 1940s have at least 50% of valid monthly data (see figure below, which has been added to the manuscript as Supplementary Figure 8). It is not the case that variability is lower in the earlier part of the time series, as shown by Supplementary Figure 9 which explicitly shows the variance in moving 30-year windows. We do note however that the 1881-1910 period was the best-observed 30-year period prior to the 1960s (86% valid months over this period). We have therefore modified our method to use this period (instead of 1911-1940, which is less well-observed) to correct the daily climatology due to mean warming since this period (see comment relating to lines “231-233” below).

lines 136-137

The "strong southward surface flow" is not clear in Figure 4. The caption indicates that colours indicate actual current speeds (but it looks to me like they indicate current speed anomalies), but there is no directional information. The arrows represent anomalies, so we cannot interpret them in terms of actual currents without knowing the reference current map.

line 147

Figure 4 is not much help here, because it shows anomalies, not actual currents.

lines 160-161

See line 147 comment.

- These three comments relate to Figure 4 which was erroneously captioned as “anomalies” when it is in fact absolute currents. This text has been corrected now. Therefore, the current speed colours do match with the current speed direction arrows indicated in the same figure. We apologize for this error and the confusion it led to.

lines 190-191

I am a bit confused as to what mechanisms are included in "air-sea heat flux" here. Presumably sensible and latent heat mechanisms and longwave radiation are included. But is shortwave radiation? That has bypassed the "air" entirely.

- Done.
- We have applied the oceanography disciplinary terminology for “air-sea heat flux” here, which represents the net contribution due to sensible and latent heat flux as well as short- and long-wave radiation (e.g. Deser et al. 2010). We apologise that this may be ambiguous to readers unfamiliar with oceanographic conventions. This has now been clarified explicitly in the “Upper ocean temperature budget” section of the Methods.

Deser, C., Alexander, M. A., Xie, S. P., & Phillips, A. S. (2010). Sea surface temperature variability: Patterns and mechanisms. *Annual review of marine science*, 2, 115-143.

lines 235-237

Why are you mentioning this, seeing as you are using the largest event, not second-largest, to set your threshold (as stated in lines 231-233 and 238-239)?

- Done.
- We apologise that this was in error. The values quoted in the text (3.1°C, 377 days) are of the event properties being attributed (the 2nd-largest event as described in the methods), not of the 2015/16 event. The text has been modified accordingly.

lines 231-233

I am not sure why you are using this reference from the distant past for estimating the threshold for the attribution calculations. If one is using an absolute (rather than return-period-based) threshold, then the need for a reference period is to remove model-observation bias. As far as the model simulations are concerned, it does not matter which period you choose: there is, by definition (ignoring drift), no bias between the historical and natural historical scenarios. But for the observations it does matter: the quality of the record is much poorer during some periods. It seems

to me that you have chosen a period with nearly the poorest quality possible within the 20th century.

- This comment raises two issues:
 - *Why use an early base period at all when using an absolute threshold?* We are using a relative threshold. Marine heatwaves are defined using the 90th percentile threshold (seasonally varying) following the Hobday et al. (2016) definition. Therefore, the choice of base period (and therefore the data from which the climatology and threshold are defined) affects what is and is not detected as a marine heatwave. We are interested in defining a marine heatwave in the modern period but relative to a climate with little anthropogenic influence, and then comparing the likelihood of those events in a natural-only climate, we must use an early base period. Otherwise we are comparing how likely an event that occurred in today's climate is due to climate change, but today's climate already includes the effect of climate change and it is critical to include this signal in the analysis.
 - *We have chosen a base period from a poorly observed period of time.* We investigated the issue of observational density further and found that the 1911-1940 period was poorly observed, relative to the later 20th century or the decades immediately preceding it. Consequently, we have now shifted our reference period to 1881-1910 (see comment regarding line "106-108" above, including the new Supplementary Figure 8) when the observational density is considerably larger, which should make our results more robust. This time period was relatively well-observed, and therefore can constrain the change in mean warming since then rather well (+0.89°C from 1881-1910 to 1982-2005), and is also early enough to have little anthropogenic climate change warming evident. This has modified the properties of the events being attributed as well as the results of the FAR analysis, all of which has been updated in the revised manuscript.

lines 242-243

What does this confidence interval represent? Is it the spread across estimates for various models?

- Done.
- This confidence interval represents the spread across estimates from various models and ensemble members based on the bootstrap resampling approach, as described in the methods. This has now been clarified in the text on this line.

line 243

The rounding error here is misrepresenting the results. FAR=-0.17 equals a Risk Ratio of 0.85, so following the number of significant digits presented in the FAR value, the 90% confidence on the RR value is consistent with a decrease, i.e. an RR<1.

- Done.
- We have increased the precision of the results to at least 2 significant digits, and the risk ratios are consistent with the FAR values at the given precision.

lines 244-245

What sort of bias might be introduced by using different periods for the two scenarios?

- The 1982-2005 period in the historical run has less influence from climate change than the 2006-2020 period in the RCP8.5 run, which is actually what we are interested in. Therefore, the 'bias' will be that 2006-2020 in RCP8.5 is 'warmer' than 1982-2005 in 'historical'. However, there is no overlap in simulation period (historical ends in 2005, RCP8.5 starts at that point) and so it is impossible to compare the two scenarios over the same time period.

lines 287-288

So was this a just a coincidence then?

- Done.
- We have investigated this further and found that a likely explanation is local adaptation at higher (cooler) latitudes. The following paragraph has been added to the “Ecological impacts” subsection:

“Interestingly, abalone mortality occurred at temperatures during the peak intensity of the MHW that were more than 7°C below the thermal maximum of 26.9°C for this species, which is found further north along southeastern Australia⁵⁵, indicating there may be local adaptation by south-east Tasmanian abalone populations to cooler maximum summer temperatures. Investigations of metabolic activity of abalone under simulated harvest stress from New South Wales and southern Tasmania populations were consistent with a hypothesis of adaptation to local thermal regimes⁵⁶. The temporal extent of the 2015/16 MHW (251 days) may have elevated background metabolic rates of abalone above normal for an extended period, reducing stored energy and consequently resilience to stress. The peak temperatures occurred at the end of the MHW, potentially creating an acute stress event for cooler adapted abalone populations, resulting in the observed mortality of wild abalone. A subsequent cut to the abalone quota for 2017 has been linked to the extra mortality as a result of this MHW.”

Ref. 56:

Moltschaniwskyj, N., Mundy, C. and Harris, J. (2014) Maximising value by reducing stress related mortality in wild harvested black-lip abalone (*Haliotis rubra*) AS-CRC Project No. 2010/704, University of Tasmania, Hobart.

line 301

Where does the "99%" value come from?

- Done.
- Added Caputi et al. (2016) as the appropriate reference.

Caputi, N., M. Kangas, A. Denham, M. Feng, A. Pearce, Y. Hetzel and A. Chandrapavan (2016). Management adaptation of invertebrate fisheries to an extreme marine heat wave event at a global warming hot spot. Ecology and Evolution: doi: 10.1002/ece3.2137.

line 308

How is that deduced?

- This is deduced from available observations and anecdotes, drawing from ecological knowledge and expertise of the co-authors. Nonetheless, we have reworded this text to reflect speculation based on local adaptation results (see response to “lines 287-288” above) rather than speculation about acute vs. chronic stress.

line 316

Do you have evidence to support "there is also a keen interest"?

- The authors have considerable experience engaging with stakeholders in the marine environment including the public, state and federal governments, and aquaculture and fisheries industries. For example, reduced tolerance to post-harvest transport and poor

condition of abalone during summer in Tasmania has led to seasonal closures for blacklip abalone (Eastern Tasmania) and greenlip abalone (North-East Tasmania), to minimise post-harvest losses (<http://dpiwwe.tas.gov.au/sea-fishing-aquaculture/commercial-fishing/abalone-fishery/abalone-closures>). The authors are well in touch with the interests around marine heatwaves in the Tasmanian and Australian contexts, and our expertise has informed this statement.

lines 323

It took me a little while to match these values to the calculations in the text, because originally they were presented to one significant digit only, whereas here you introduce more precision.

- Done.
- We have increased the precision of the results to at least 2 significant digits (as per comment above).

lines 430 and 433

Do shortwave radiation fit in "air-sea heat flux". It is not a flux of heat between the two, but then I do not see where else it would go.

- Done.
- See comment above on the standard terminology in the discipline of oceanography for “air-sea heat flux”, and the clarifications added to the text in the Methods.

line 471

I think you mean "particularly for the historical experiment", because that has fewer simulations in your table.

- Our original statement is correct. As stated, the historicalNat experiment was the limiting one in terms of the number of CMIP5 models available with daily SSTs (‘tos’ variable).

lines 489-490

If I understood correctly, you are using an absolute threshold, which also means that the frequency of exceedence will be affected by relative bias in the mean ("a"), trend ("bt"), and annual cycle (" T_t^S ") components. Did you account for any of these as well? It does not sound like it here.

- No, a relative threshold is used based on percentiles. Marine heatwaves (MHWs) are defined as being above the 90th percentile (seasonally varying) and therefore the offset in the mean (a) is not relevant. The only way in which the mean is relevant is the difference in the mean from the historical to the RCP8.5 scenarios, since the historical climatology is used to detect RCP8.5 MHWs. This also holds for the seasonal cycle (T_t^S). Since the threshold is defined as seasonally-varying, we are not concerned with biases against the observations in the seasonal cycle itself. In other words, when the MHWs are detected, the mean and seasonal cycle are irrelevant – only the non-seasonal variability defines how the threshold sits above the climatological mean. Regarding the trend (bt), in essence this is the primary signal which the climate models are providing so we would not want to correct for any perceived error in it. Effectively, we are using the climate models to provide us with a good estimate of the trend.

lines 493-506

As above, I am not clear on why you needed the extra steps, which amongst other things introduce sampling variance and hence broaden the uncertainty on your results. Why not just use the well-

observed 1982-2005 period and, for all scenarios, use the data from the historical 1982-2005 data for the calibration?

- This is the same issue as raised above (comment regarding “lines 231-233”), and so we here defer to the response provided above.

lines 507-509

This could introduce a bit of a problem if models respond differently, because you have inconsistently different sample sizes across scenarios. For instance, the estimate for the 1982-2005 period are only half-informed by the CNRM-CM5 model (nearly no historical simulations, five historicalNat simulations) but are well informed in the estimate for the 2006-2020 period. Similarly, for 1982-2005 you are essentially estimating historicalNat in part from CanESM2 and CNRM-CM5, but not for historical. If you are calling these "PDFs", then this sampling represents an unobvious distribution on your prior. There are a number of possible "uniform-like" priors, most of which would involve weighing models by the inverse of their sample size for a given scenario. In any case, this choice of prior needs to be justified.

- Done.
- We have now weighted the ensemble members by the inverse of the number of ensemble members for that particular model, thereby weighting equally across models rather than across ensemble members irrespective of which model. This has been indicated in the methods section, and the results have been updated accordingly (the results were not sensitive to this change).

lines 515-517

This agrees with lines 235-237, but contradicts the lines immediately preceding and following 235-237.

- Done.
- We apologise that the text in lines 235-237 was incorrect, and has now been corrected accordingly (see response to comment regarding “lines 235-237” above).

lines 532-533

Which means your estimated uncertainty range are probably about $\sqrt{2}$ larger than the uncertainty around the estimates value?

- We very much appreciate this comment, but unfortunately have been unable to interpret further what this means without further clarification from the reviewer. We would be more than happy to consider and address this with further clarification.

lines 536-537

Um, this is manipulating results. First, if this occurs less than 5% of the time it will not affect your estimate of the 90% confidence interval. Second, if that possibility is within the resolution of your frequency estimates, then that is the result. If we know that a 2015/2016-like event occurred in a historicalNat-like boundary condition in the real world, then we can state that a lower bound of 0/N frequency is impossible, but we cannot automatically infer than an actual frequency of less than 1/N is also impossible.

- Done.
- We have no longer removed bootstrap samples with FAR values of 1, and removed these lines from the text. Note that this has modified the results somewhat, and the results have been updated throughout the main text. The most notable change is a dramatic increase in

the FAR estimates on MHW duration, since events of this duration were so rare in the historicalNat simulations.

- We have also replaced the 90% confidence interval (two-tailed “very likely” statement) with the 10th percentile as a lower 10% confidence interval (one-tailed “very likely” statement) which is more in line with the existing methods presented in the literature (e.g. Lewis and Karoly, 2013, GRL, 40:3705-3709).

lines 540-542

So actually your historicalNat simulations are not a sample of a natural climate, but rather an extended sample of a 1910-1940-like climate with added variance from the extra bias correction steps required?

- Yes, this is true. Our “natural climate” is not purely a simulated “natural / pre-industrial” climate but rather the anomalies of such a simulate climate relative to its own 1911-1940 climatology. We are not the first study to perform this type of analysis as this has also been done recently, e.g., Lewis and Karoly (2013).

Lewis, S. C., & Karoly, D. J. (2013). Anthropogenic contributions to Australia's record summer temperatures of 2013. *Geophysical Research Letters*, 40(14), 3705-3709.

lines 728-729

For NOAA OI or HadISST?

- Done.
- For NOAA OI SST, this has been clarified in the figure caption.

Figures 3, 4, and 5

Anomalies from what reference period?

- Done.
- Figure 3 anomalies are relative to the 1982-2005 base period, now indicated in the figure caption.
- Figure 4 does not present anomalies (as per comment “lines 136-137” above).
- Figure 5 anomalies are relative to the 2012-2016 period (all available CFSv2 data), now indicated in the figure caption.

line 750

The current speed (colours) and the current speed anomalies (arrow lengths) look rather similar. Are the colours actually showing anomalies too?

- As indicated above, Figure 4 does not present anomalies but absolute currents – now clarified in the figure caption. This clarification should now make the arrows and colours much easier to interpret.

line 770

These triangles are very hard to see.

- Done.
- Triangles made larger and moved to a location which makes them easier to identify.

Supp line 47

Add quotation marks to "Complete".

- Done.

*Sincerely,
Daithi Stone*

Response to Reviewer #3

Review of “The unprecedented 2015/16 Tasman Sea marine heatwave” by Oliver et al.

This manuscript discusses an extreme temperature anomaly during 2015-2016 in the Tasman Sea. The paper, as the authors noted is a systematic documentation, which discusses the evolution, causal factors, impacts, and future reoccurrence of the extreme event. The major conclusion of this paper is that the southward advection of warm water was the dominant driver of the extreme event and the likelihoods of this kind of event will increase due to anthropogenic influence. The authors claim that this paper is “an advancement” of previous studies, which focused on different aspects (physical drivers, ecological impacts, and the role of anthropogenic forcing) of similar extreme events occurred in other parts of the world’s ocean. I have reservations about this statement, because the scientific methods in this paper are mostly, if not all, based upon these previous studies and piecing together different aspects of a problem doesn’t necessarily mean scientific advancement. Moreover, trying to cover every aspect of the problem in a short contribution might instead dilute the core science. With that said, this is a timely report on an increasingly important issue, which should be of interest to a diverse readership. I have some comments below to help the improvement of this manuscript.

- Done.
- We acknowledge and, on reflection, agree with the reviewer’s comment. We have endeavoured to shift the tone of the abstract and introduction to better reflect the nature of the work: i.e. that it combines the findings from analyses of several factors associated with this extreme marine heatwave event.
- We have added the following line to the abstract: “We report on several inter-related aspects of this event: observed characteristics, physical drivers, ecological impacts, and the role of climate change.”
- The third paragraph of the introduction has been revised as: “This paper discusses the 2015/16 Tasman Sea marine heatwave from observations and ocean models, diagnoses its physical drivers and the role of anthropogenic climate change, and describes the ecological impacts that occurred. By collating the analyses together we can integrate the inter-related consequences of this extreme event from the physical drivers and climate change, and their impacts on marine ecosystems. Thus we illustrate how to characterise marine heat waves regionally, where there is a growing need for a clear and timely analyses for such events.”
- In the last paragraph of the discussion, we have removed the original statement that the study represents an “advancement over previous studies”. We have now revised this statement to read “This is in contrast to previous studies which have separately examined events’ physical drivers³⁻⁷, ecological impacts⁷⁻¹¹ or the role of climate change⁹⁻¹²”.
- [See also Reviewer #2, Comment #1]

Major comments:

1. Perhaps the most exciting scientific aspect of this paper is the investigation of the physical drivers of the extreme event. Looking at Figure 3 to Figure 6, I generally agree with the authors that ocean advection should be the primary driver. However, the key statement in the abstract “the warming was dominated by anomalous net southward advection linked to the East Australian Current” was not well supported. In Figure 6, TH is the total horizontal advective flux that contains advective fluxes from both directions and doesn’t directly indicate the role of southward advection of the western boundary current. Figure 4 shows some southward surface geostrophic current anomalies, but doesn’t directly show the depth-averaged (0-100m) heat divergence anomaly, which should provide direct evidence to support the statement of southward advection. My suggestions

would be to decompose the total advection term in Figure 6 to quantify the contributions in both directions, and/or to plot depth-averaged heat divergence anomaly.

- Done.
- This is a very good point, which we have explored leading to one new result, now added to the paper. We have decomposed the horizontal advection component into four sub-components, that is to say the temperature budget contributions from input across the north, south, west and east faces of the study region (the SEAus box). This can be seen in the Figure below, presented in the same format as the temperature budget figure in the originally submitted manuscript (i.e. Fig. 6; that is to say (a) a climatology, (b) the 2015/16 year alone, and (c) the 2015/16 year anomalies). The terms plotted represent the contribution to the time-integrated temperature tendency from the horizontal faces. On this new figure, positive (negative) values indicate an input (output) of temperature to the SEAus box across the indicated face. In 2015/16 there was anomalously strong input across the north face (solid blue line, panel c). There was anomalously strong output across the west face (solid red line) but not enough to compensate for the input across the north face. The anomalous input/output across the south (dashed blue line) and east (dashed red line) were generally in near-balance throughout 2015/16. We can associate the input across the north face with the East Australian Current (EAC) Extension and this supports our claim that the anomalous forcing was due to an EAC Extension event. It is interesting that output is primarily to the west rather than south, indicating a dominant role for the Tasman Outflow rather than direct southward advection into the Southern Ocean.

- This figure has been added to the manuscript (Supplementary Figure 7) and an associated paragraph discussion around this result has been added to the “Driving mechanisms” subsection in “Results”.

2. *The other important discussion of this paper is on the future reoccurrence of extreme temperature anomaly like 2015/2016. This was discussed from page 10-14. However, I found the discussions confusing in that the numbers in the text were not obviously supported by the figures. For example, in the key conclusion “an extreme warming event of this duration was 8.5 times as likely to occur due to anthropogenic climate change; and event of this intensity was 5.9 times as likely”, the numbers 8.5 and 5.9 were not discussed/explained in the text. I have listed some other similar issues below. This part would be much improved if the authors can be explicit about the derivation of the numbers.*

- Done.
- We apologise that this was a source of confusion for two reasons, which have now been addressed:
 - First, there was an inconsistency in the precision of the numbers presented, i.e. occasionally they were presented to two significant digits (5.9 times) and occasionally to one significant digit (6 times). Now all instances of these values are presented to two significant digits.
 - Second, the numbers were not indicated on the supporting figure (Figure 7). As described below (comment #5), we have now indicated the median and confidence bound FAR values on the figure appropriately – so the values quoted in the text can be easily identified on the figure.

Other comments/corrections:

1. *I would always be cautious about using the word “unprecedented”. The fact that the magnitude of the anomaly was the largest on record doesn’t mean it is unprecedented.*

- The Oxford English Dictionary defines ‘unprecedented’ as ‘Having no precedent; unparalleled; never previously done, known, or experienced.’ We can certainly say that an event of this magnitude in this region had not previously been observed, within the satellite observational record. However, we recognise that we cannot know whether or not an event like this has ever occurred since the record length of satellite observations is only 35 years. Our use of the word ‘unprecedented’ in the manuscript is in reference to precedent in the satellite SST record. This usage is consistent with other recent literature, for example ‘unprecedented’ was used to describe the 2011 marine heatwave off Western Australia (Wernberg et al. 2013).

Wernberg, T., Smale, D. A., Tuya, F., Thomsen, M. S., Langlois, T. J., De Bettignies, T., ... & Rousseaux, C. S. (2013). An extreme climatic event alters marine ecosystem structure in a global biodiversity hotspot. *Nature Climate Change*, 3(1), 78-82.

2. *Page 6, line 107: “top 10 greatest” was not obvious from the Figure. Marking the top 10 months in the figure would help.*

- Done.
- These months are now indicated by red-filled dots in Figure 1, panels (b) and (c) and corresponding text has been added to this line in the manuscript.

3. Page 6, lines 111-125: Any specific reason why the emphasis was placed on Maria Island mooring?

- Maria Island is an Integrated Marine Observing System (IMOS) National Reference Station (NRS) and is therefore a committed long-term ocean monitoring site, including quality control of the data (<http://imos.org.au/nrs.html>). It has also been the site of noted previous studies of long-term warming in the region, using a now-discontinued hand-collected record going back to the mid-20th century (e.g. Holbrook and Bindoff 1997; Ridgway 2007; Johnson et al. 2011). Therefore, it plays a significant role in the literature and amongst researchers as an iconic and robust measure of oceanic change off southeastern Australia and in the Tasman Sea. For those reasons we emphasised the Maria Island results, and complemented them with data from the other loggers around Tasmania for further context. Note that since original submission, the large data gap in the Maria Island record during the 2015/16 MHW has been filled and the manuscript has been updated accordingly with these new data (Fig. 2 and results associated with the Maria Island MHW properties).

Holbrook, N. J. and N. L. Bindoff (1997). Interannual and decadal temperature variability in the southwest Pacific Ocean between 1955 and 1988. *Journal of Climate*, 10, 1035-1049.

Johnson, C. R. et al. (2011). Climate change cascades: Shifts in oceanography, species' ranges and subtidal marine community dynamics in eastern Tasmania. *Journal of Experimental Marine Biology and Ecology*, 400(1), 17-32.

Ridgway, K. R. (2007). Long-term trend and decadal variability of the southward penetration of the East Australian Current. *Geophysical Research Letters*, 34(13).

4. Page 11, lines 240-243: “The best estimate (i.e., median) FAR value ... to be 0.58, which based on Intergovernmental Panel on Climate Change (IPCC) ... is ‘very likely’ ... to be between -0.17 and 0.84 ...”. This sentence reads awkward. 0.58 is very likely to be between -0.17 and 0.84?

- Done.
- This has been reworded to remove ambiguity in the sentence.

5. Page 11, line 242-244: The numbers -0.17, 0.84, 1, 6, and 2 are not clearly supported by the figures.

- Done.
- The best estimate and confidence bound FAR values for the 1982-2005 and 2006-2020 periods are now indicated in Figure 7 by filled and open circles respectively.

6. Page 11, line 244-245: “...historical simulation period 1982-2005 compared to the historicalNat simulations over the period 1850-2005”. Are the two periods directly comparable?

- This is standard practice (Lewis and Karoly 2013, King et al. 2015). “HistoricalNat” is a stationary climate (natural variability notwithstanding) and so we chose a long time period (1850-2005) to get the best statistics possible. “Historical” is a nonstationary climate due to anthropogenic influences and so different, shorter periods will provide different statistics, and so we chose a recent period (1982-2005) as representative of now, or at least the very recent past.

Lewis, S. C., & Karoly, D. J. (2013). Anthropogenic contributions to Australia's record summer temperatures of 2013. *Geophysical Research Letters*, 40(14), 3705-3709.

King, A. D., van Oldenborgh, G. J., Karoly, D. J., Lewis, S. C., & Cullen, H. (2015). Attribution of the record high Central England temperature of 2014 to anthropogenic influences. *Environmental Research Letters*, 10(5), 054002.

7. *Page 12: Again, numbers were not clearly derived or marked in the figures.*

- Done.
- See comment #5 above.

8. *Page 20, line 453 and Figure 6: Why only use 3 years to construct the climatology?*

- Done.
- Only those years were available for the OceanMAPS data product. This has now been clarified in the Methods text on the specified line.

REVIEWERS' COMMENTS:

Reviewer #1 (Remarks to the Author):

I am happy with the modifications that were done to the MS and have no further questions or comments to add.

Reviewer #2 (Remarks to the Author):

The authors have addressed my concerns with the original manuscript and I now recommend publication in Nature Communications.

Reviewer #3 (Remarks to the Author):

I am glad to see that the authors have addressed my concerns and suggestions. The evidence supporting one of the major claims on southward advective flux has been added and thus the section of physical drivers has been strengthened. The discussions of future reoccurrence have also been clarified with consistent numbers and improved figures. Other minor issues in the text and figures have also been resolved. Now the manuscript reads much better and the statements are more solid. Although I am still not sure about the usages of "unprecedented", I understand the authors' standing point and the eye-catching need for this type of article. Now I recommend the publication of the manuscript in Nature Communications.